# PhysTTT: Accurate and Lightweight Cross-Domain Heart Rate Measurement with Test-Time Training

## Abstract

Remote photoplethysmography (rPPG), a contactless technology for measuring physiological signals, holds significant promise for smart healthcare and affective computing. However, a key challenge for existing deep learning methods is the paradox between maintaining high measurement accuracy and ensuring low computational cost, especially in cross-domain scenarios. To address this, we propose PhysTTT, a novel and lightweight framework for heart rate measurement that integrates multiple 1D-CNNs with residual structures and a Test-Time Training (TTT) layer. Multi-time frame differences fusion and 1D-CNNs extract spatio-temporal features from facial video sequences by modeling subtle brightness variations, the TTT layer compresses the context information into a learnable vector space, enhancing the temporal modeling capability. Crucially, the TTT mechanism enables the model to adapt to unseen data distributions during inference, significantly boosting cross-domain generalization. Extensive experiments demonstrate that PhysTTT achieves state-of-the-art accuracy in both in-domain and cross-domain evaluations, offering an optimal balance of high performance, strong generalization, and low computational cost. Our code is publicly available at https://anonymous.4open.science/r/PhysTTT-B605/.

## 1 Introduction

Blood volume pulse (BVP) signal, which arises from pulsatile changes in blood volume with each heartbeat, provides a critical foundation for extracting vital physiological indicators such as heart rate (HR), respiration rate (RR), and heart rate variability (HRV). These indicators are highly valuable for medical diagnosis, health monitoring, and affective computing. Photoplethysmography (PPG) is a conventional method for capturing the BVP signal by measuring light absorption and scattering changes in tissue using specialized contact sensors. However, the requirement for physical contact limits its convenience and continuous use. In contrast, remote photoplethysmography (rPPG) is a non-contact alternative that uses a common camera to detect subtle, heartbeat-induced skin color variations in facial or skin regions. By analyzing these periodic fluctuations, rPPG enables the inference of the BVP signal and subsequent physiological parameters. This approach eliminates the need for specialized wearable equipment, offering significant advantages in terms of convenience, low cost, and ease of deployment, thereby holding broad application prospects.

Early unsupervised methods (Poh et al., 2010); (Poh et al., 2011); (De Haan & Jeanne, 2013); for rPPG signal extraction relied on statistical approaches, operating on the principle that subtle periodic skin color changes in video correspond to the PPG signal. These techniques combined different region-of-interest (ROI) selections (Lam & Kuno, 2015) with color space projections (Wang et al., 2016); (Tulyakov et al., 2016) to recover rPPG signals from static videos. However, the extracted signals are often weak and highly susceptible to ambient lighting variations, motion artifacts, and background noise, making accurate BVP signal recovery in complex real-world scenarios challenging with traditional signal processing. The development of deep learning prompts a paradigm shift towards data-driven supervised methods. Early models utilized 2D-CNNs (Chen & McDuff, 2018); (Niu et al., 2020); (Liu et al., 2020); (Narayanswamy et al., 2023) and 3D-CNNs (Yu et al., 2019); (Zhao et al., 2021); (Yue et al., 2023); (Sun & Li, 2024) to extract spatio-temporal features, improving robustness to motion artifacts. This field subsequently transitioned to the use of Transformer

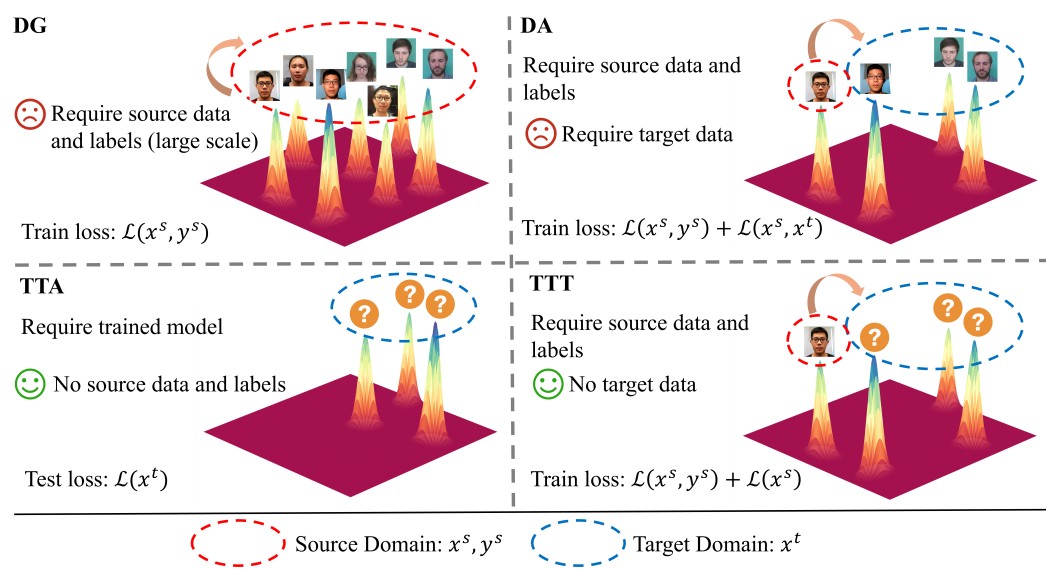

Figure 1: Illustration of different cross-domain methods. Domain generalization (DG) requires a large scale source data. Domain adaptation (DA) requires the data and distribution of the target domain as guidance during inference. Test-time adaptation (TTA) aims to fine-tune a pre-trained source model during inference time without accessing source domain. Test-Time Training (TTT) can adapt to the data distribution of the target domain by dynamically updating the parameters of the hidden state during the testing time.

architectures (Yu et al., 2022); (Liu et al., 2023a); (Liu et al., 2025) and Mamba-based methods (Luo et al., 2024); (Yan et al., 2025) (Zou et al., 2025b), which aim to better capture long-range temporal dependencies and efficiently model the quasi-periodic nature of physiological signals with linear computational complexity.

Despite these advances, significant challenges remain. Transformers are often computationally expensive and parameter-heavy. While State-Space Models (SSMs) offer a better balance between linear complexity and long-range modeling (Gu et al., 2021), SSM is usually modeled as a linear time-invariant system, which may cause the model to lack sufficient flexibility to adapt to new patterns. Mamba has significantly enhanced its generalization ability in fields such as language by introducing a selective mechanism. However, this mechanism disrupts the linear time invariance attribute, making it impossible to train with efficient convolutional patterns anymore. (Gu & Dao, 2024); (Lieber et al., 2024). Domain generalization (DG)-based methods (Lu et al., 2023); (Sun et al., 2023) seek to enhance inherent model generalizability without target domain data but often require large, multi-domain training sets and substantial parameters. Domain adaptation (DA)-based (Du et al., 2023) approaches leverage target domain data to improve adaptability but are impractical when potential target domains are unseen or distributions shift significantly, leading to performance degradation. Unlike DG and DA, test-time adaptation (TTA) (Wang et al., 2021); (Niu et al., 2023); (Li et al., 2024); (Huang et al., 2026) aims to adjust a pre-trained source model during inference by learning from the unlabeled target data, without accessing the distributions and labels of both source and target data.

To improve the generalization and applicability of the model in various rPPG measurement scenarios, we propose a new heart rate measurement framework, based on the Test-Time Training (TTT) layer, which is a new sequence modeling layer with linear complexity and expressive hidden states. Firstly, the differential frames extracted from the face video are fed into the multi-layer 1D-CNN for further deep feature extraction. Then, the training process is divided into two nested loops. Design loss functions in the outer loop to learn the inherent physiological characteristics of the BVP signal from the perspectives of trend, frequency and peak position. The inner loop is the TTT layer. Each hidden state of this sequence modeling layer is designed as a machine learning model to better store the context information of the input sequence and is updated through a self-supervised learning

strategy. This method can also continuously update the model weights during testing to adapt to the real data distribution of the target domain, thereby enhancing the cross-domain capability. The main contributions are as follows:

- We propose PhysTTT, an accurate and lightweight cross-domain heart rate measurement framework, which compresses the context information of video frame sequences into a self-supervised learning model through the TTT layer, and at test time learning data distributions in unseen domains. To the best of our knowledge, this is the first attempt of a new paradigm of training at test time in the rPPG research, thus cracking the intractable problem of not being able to predict and adapt the real data distribution of different future application domains at training time.
- In order to more realistically fit the waveform characteristics of the BVP signal, we align the features of the predicted signal and ground truth signal from multiple dimensions to achieve fine-grained waveform recovery.
- Extensive experiments with in-domains and cross-domains show that the proposed method outperforms previous state-of-the-art methods, demonstrating its accurate and efficient measurement capability and stability across diverse domains.

## 2    RELATED WORK

### 2.1    RPPG MEASUREMENT

Early rPPG researches primarily employed unsupervised learning to recover signals by statistically combining features from different color channels (De Haan & Jeanne, 2013); (Tulyakov et al., 2016) or fusing information from multiple ROIs (Li et al., 2014); (Lam & Kuno, 2015). The field subsequently witnessed a paradigm shift with the advent of deep learning, which leveraged neural networks' powerful spatio-temporal representation capabilities. Spatio-temporal map methods utilizing facial ROIs became prominent, employing 2D-CNNs (Chen & McDuff, 2018); (Niu et al., 2020); (Liu et al., 2020) and 3D-CNNs (Yu et al., 2019); (Zhao et al., 2021) to extract rPPG features, thereby mitigating interference from non-skin regions and incorporating temporal information. To further enhance long-range dependency modeling, Transformer-based approaches (Yu et al., 2022) are introduced, however, their high computational complexity often hinders deployment on resource-limited devices. Mamba-based methods (Luo et al., 2024); (Zou et al., 2025b); (Yan et al., 2025) can effectively capture long-distance dependencies and quasi-periodic patterns in physiological signals with linear complexity. Despite these advantages, SSM-based methods still face challenges in generalization, parallel processing, and maintaining robustness in unseen scenarios.

### 2.2    TEST-TIME TRAINING LAYER

The TTT layer is an emerging sequence modeling layer that uses self-supervised learning to compress historical context information into hidden states and continuously update model weights during test-time to adapt to the actual data distribution of future target domain, thereby enhancing cross-domain capabilities (Sun et al., 2019); (Sun et al., 2024). Compared with Transformers with secondary complexity attention and RNNs and their variants with limited expressive power and hidden states (Kaplan et al., 2020), the TTT layer has linear complexity and expressive hidden states. Compared with TTA, TTT performs a complete gradient update during testing, which can significantly adjust the parameters of the hidden state and has a stronger adaptability to distribution offsets. Unlike other computer vision tasks, signal extraction based on rPPG relies on subtle facial color changes. Most TTA methods are targeted at classification tasks and are not suitable for regression tasks like rPPG in themselves because video frames arrive in a continuous and sequential manner, which leads to a lack of appropriate supervision. Moreover, some instantiation-level TTA methods may apply additional auxiliary tasks (D'Innocente et al., 2020) or rely on specialized network architectures (Klingner et al., 2022) during the pre-training and adaptation process, which leads to a narrowing of the application scope of TTA and an increase in computational costs. A series of studies have shown that the TTT layer demonstrates advantages in the fields of natural language processing, large language models, and generative artificial intelligence (Dalal et al., 2025). Inspired by this, we explore the application potential of the TTT layer in rPPG measurement to enhance the cross-domain generalization performance of the model.

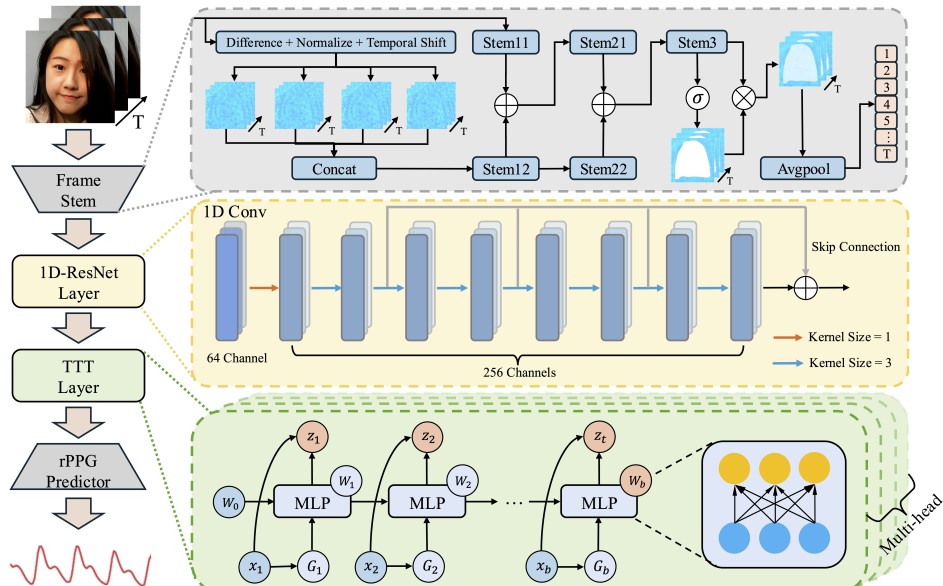

Figure 2: The overall network architecture of PhysTTT. It consists of frame stem, 1D-ResNet layer, TTT layer, and rPPG predictor head. Where "+" represents addition, "×" represents multiplication, "$\sigma$" represents the sigmoid function, $x_1, x_2, \cdots, x_b$ represent the input sequence per batch, $G_1, G_2, \cdots, G_b$ represent the descent direction, $W_0, W_1, W_2, \cdots, W_b$ represent the weights of MLPs.

## 3 METHOD

### 3.1 NETWORK ARCHITECTURE

As shown in Figure 2, the overall framework of PhysTTT comprises four main components: frame stem, 1D-ResNet layers, TTT layer, and rPPG predictor. The frame stem first processes the video stream by cropping facial ROIs and extracting features between frames, following a multi-frame fusion strategy (Zou et al., 2025a). This pre-processing step amplifies subtle skin color variations and guides the network to focus on BVP-related signals within skin pixels. The resulting feature sequence is then fed through multiple 1D-CNN layers with residual connections, which maintain a constant output channel dimension to extract local spatio-temporal information. Subsequently, the TTT layer performs temporal modeling by capturing dependencies across frames and enhancing feature interactions, thereby facilitating the extraction of quasi-periodic BVP patterns. Finally, the predictor estimates the BVP signal waveform, from which the heart rate is derived.

### 3.2 FRAME STEM

In the rPPG task, extracting inter-frame differences has been proven to be helpful for robust BVP signal recovery (Chen & McDuff, 2018). We apply the fusion stem (Zou et al., 2025a), which integrates frame differences into the original frames, achieving frame-level representation perception of BVP wave variations. Specifically, for the face video stream $X \in \mathbb{R}^{T \times 3 \times H \times W}$, where $T$, 3, $H$, $W$ represent the sequence length of the input video frames, the number of color channels, and the height and width of the image, respectively. Firstly, the differences of the two adjacent time steps of $X_t$ are calculated successively and standardized to obtain $D_k$. They are concatenated and fed into different 2D convolutional stems for feature extraction and fusion.

$$D_k = |X_{k+1} - X_k|, \; k \in \{t-2, t-1, t, t+1\}, \quad (1)$$

$$X_{diff} = Concat(D_{t-2}, D_{t-1}, D_t, D_{t+1}). \quad (2)$$

Meanwhile, the original frames are also synchronized for feature extraction through the residual structure.

$$X_{Stem1} = Stem11(X) + Stem12(X_{diff}). \quad (3)$$

Further, the fused $X_{Stem1}$ and the features processed by Stem12 in the previous step are fed into Stem21 and Stem22 for further feature fusion, and the two outputs are summed up to obtain $X_{Stem2}$, where the height and width of a single-frame differential image of a face are compressed into $H/8$ and $W/8$. Next, $X_{Stem2}$ is input into Stem3, a 2D convolutional layer with a $3 \times 3$ convolutional kernel, to integrate the spatial information into the channel. Finally, the height and width of the image are averaged to obtain the final feature $X_{Stem} \in \mathbb{R}^{T \times C}$, where $C$ represents the final number of channels.

$$X_{Stem3} = Stem3(X_{Stem2}), \ mask = \frac{(H/8 \times W/8 \times \sigma(X_{Stem3}))}{(2\sum_1^W \sum_1^H \sigma(X_{Stem3}))}, \tag{4}$$

$$X_{Stem} = Avg(X_{Stem3} \times mask). \tag{5}$$

### 3.3   1D-RESNET

Many recent self-supervised methods or applications still use CNN architectures as backbone networks or encoders (Lu et al., 2021); (Joshi et al., 2024); (Joshi & Cho, 2024). Zhang et al. (2025) applies ResNet as an encoder by integrating explicit and implicit prior knowledge to improve the generalization performance of the model. Although the frame stem has initially extracted the inter-frame differences, the two global average pooling of the $H$ and $W$ channels at the stem end may lose some information. Therefore, we utilize multiple 1D-CNNs in order to deeply focus on the rPPG representation of each differential frame. Specifically, the fused input $X_{Stem} \in \mathbb{R}^{T \times C}$, first converts the channel space $C$ from 64 to 256 by an initial 1D-CNN with a convolutional kernel of 1 and maintains a fixed number of channels of 256 in all subsequent 1D-CNN layers, which have the same kernel size, stride, and padding, which are 3, 1, and 0. This design choice helps to reduce the computational complexity because increasing the number of channels significantly increases the computation time. At the same time, the output of every second convolutional layer is summed up with the output of the last convolutional layer through skip connections, which helps to maintain the original feature information, and the coordinated update of gradients in different paths improves the robustness of the model and makes it easier to train the network.

### 3.4   SELF-SUPERVISED STRATEGY IN TTT LAYER

As shown in Figure 3, the RNN layer's ability to memorize long contexts is limited by the amount of information that can be stored in its hidden state due to its structure, which makes it difficult to capture long-distance dependency and suffers from the problem of exploding or disappearing gradients. Most TTA algorithms are primarily designed for classification tasks, leveraging entropy-based measures in normalization layers or pseudo-labeling techniques. However, these strategies are inherently unsuitable for regression tasks such as rPPG. The architecture of the TTT layer resembles that of a RNN, demonstrating its ability to model time series. Moreover, it replaces the original hidden states in RNN with machine learning models, eliminating the performance limitations of the RNN layer in long contexts due to the expression of its fixed-size hidden states. This structure with stronger generalization ability also expands its modeling capability for long time series coherence and periodic data. Meanwhile, the parameters of these hidden states can be dynamically adjusted according to the target domain, enabling adaptation to previously unseen data distributions. We consider training a larger network as an outer loop and training the model parameters within the TTT layer as an inner loop. The outer loop aims to optimize the initial parameters of the network and learn a general hidden state update strategy, enabling the inner loop to effectively update the state through self-supervised tasks. The inner loop utilizes the strategies learned from the outer loop to optimize the hidden state during testing through self-supervised learning, thereby enhancing the adaptability to the current test sequence.

#### 3.4.1   OUTER LOOP

Macroscopically, the rPPG task maps input image sequences to compressed, high-dimensional feature vectors through a series of feature extraction steps. A TTT layer then performs temporal modeling, yielding time-synchronized rPPG signal representations via dedicated prediction heads. The outer loop task is therefore formulated as minimizing the discrepancy between the predicted and ground truth signals. To precisely align the predicted and actual BVP waveforms, ensure multi-

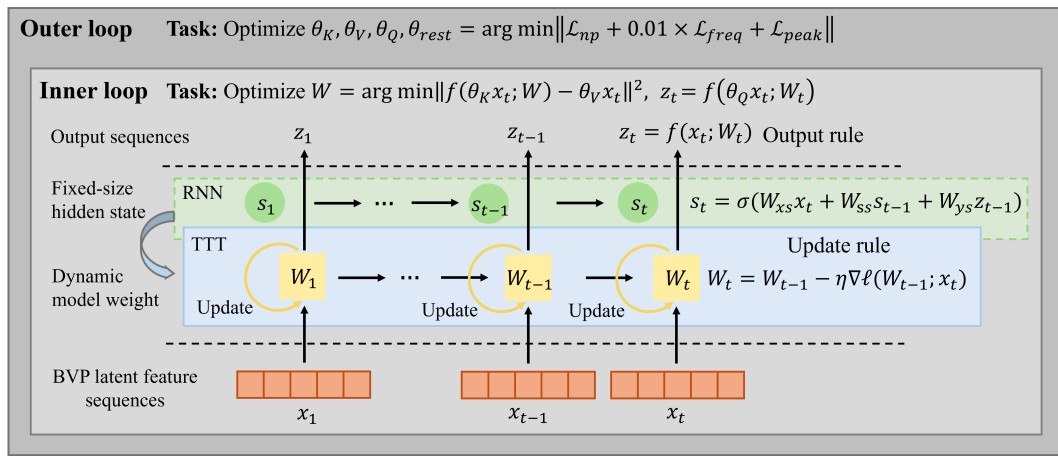

Figure 3: An overview of the parameter update process in rPPG tasks. The RNN layer compresses the context information into a fixed-size hidden state, while the TTT layer can dynamically update the model's parameters to adapt to the data distribution in unseen domains.

ple features consistency, and robustly reconstruct the inherently weak BVP signal, the outer loop objective requires the following three components.

**Trend Pattern Alignment.** The preservation of temporal trends is crucial in time-series prediction, even when signal amplitude and phase vary. This is particularly important for rPPG, given the strong quasi-periodic nature of PPG signals. Therefore, incorporating a loss component that maintains correlation is highly beneficial. Accordingly, the negative Pearson correlation loss $\mathcal{L}_{np}$ is defined to quantify the dissimilarity between the predicted signal $\hat{y}_t$ and the ground truth PPG signal $y_t$.

$$\mathcal{L}_{np} = 1 - \frac{T\sum_{t=1}^{T}\hat{y}_t y_t - \sum_{t=1}^{T}\hat{y}_t\sum_{t=1}^{T}y_t}{\sqrt{T\sum_{t=1}^{T}\hat{y}_t^2 - \left(\sum_{t=1}^{T}\hat{y}_t\right)^2}\sqrt{T\sum_{t=1}^{T}y_t^2 - \left(\sum_{t=1}^{T}y_t\right)^2}}, \quad (6)$$

where $T$ represents the time length of the input signal.

**Frequency Domain Alignment.** Motivated by the strong temporal dependence of rPPG signals, we decompose them via Fourier analysis to access frequency components critical for estimating heart rate and its variability. Aligning the dominant spectral features of the predicted and ground truth signals suppresses irrelevant high- and low-frequency noise while enhancing sensitivity to physiological patterns. This is achieved through a power spectral density loss $\mathcal{L}_{freq}$ computed as the error between $\hat{y}_t$ and $y_t$ over the 45–150 beats per minute (BPM) range.

$$PSD(f) = \left|\int_{-\infty}^{\infty}\hat{y}(t)e^{-j2\pi ft}dt\right|^2, \quad (7)$$

$$\{f_1, f_2, \cdots, f_k\} = argTopk\left(PSD(f)\right), \ k \in \{1, 2, \cdots, T/2\}, \quad (8)$$

$$\mathcal{L}_{freq} = \frac{1}{F}\sum_{k=1}^{F=T/2}|\hat{f}_k - f_k|, \quad (9)$$

where $PSD(f)$ denotes the power spectral density (PSD) of the input signal, $f_k$ represents the frequency index corresponding to the $k$-th PSD value, and $F$ signifies the number of discrete frequency components within the typical heart rate range.

**Waveform Feature Alignment.** Beyond coarse-grained alignment within the standard heart rate range, fine-grained alignment of pulse wave peaks is critical for heart rate measurement, as it enables precise localization of instantaneous pulse events corresponding to cardiac contractions. We apply a peak detection algorithm to the predicted BVP signal to identify the primary peak positions. The peak alignment loss $\mathcal{L}_{peak}$ is consequently defined as the discrepancy between the timestamps of

the identified peaks in $\hat{y}_t$ and those in $y_t$.

$$\mathcal{L}_{peak} = \frac{1}{T}\sum_{t=1}^{T}\alpha\left|1 - \frac{PD(\hat{y}_t)}{PD(y_t)}\right| + (1-\alpha)\sum_{k=1}^{K}|t_k(PD(\hat{y}_t)) - t_k(PD(y_t))|, \qquad (10)$$

where $PD(\cdot)$ represents the amplitude detected by peak detection, $t_k(\cdot)$ indicates the index of this peak in the time series, $\alpha$ represents the balance factor, and $K$ represents the total number of all detected peaks.

Finally, the loss of the outer loop can be represented as:

$$\mathcal{L}_{outer} = \mathcal{L}_{np} + 0.01 \times \mathcal{L}_{freq} + \mathcal{L}_{peak}. \qquad (11)$$

### 3.4.2 INNER LOOP

Self-supervised learning facilitates the encoding of extensive training data into model parameters. Our TTT layer employs an end-to-end strategy, optimizing directly for the prediction target $z_t$. The choice of self-supervised task determines the features learned by the parameters $W$. We treat the context as an unlabeled dataset, where each hidden state $s_t$ (derived from the latent BVP feature $x_t$) acts as a model tasked with predicting the rPPG-relevant target $z_t$. Here, the model $f$ is a two-layer perceptron. The hidden state $s_t$ encapsulates the model's weights $W_t$ at time $t$, and the output $z_t$ is the prediction for $x_t$ made using these weights. The update rule involves one step of gradient descent with rate $\eta$ on the self-supervised loss $\ell$:

$$W_t = W_{t-1} - \eta\nabla\ell(W_{t-1}; x_t), \ z_t = f(x_t; W_t). \qquad (12)$$

To avoid trivial solutions and promote the learning of domain-generalizable features in $z_t$, we define the inner loss $\ell$ as a reconstruction task for $x_t$. This objective discourages overfitting to the source domain by acting as an implicit gate, forcing the model to discard non-essential source-specific information when approximating the target domain distribution. The input $x_t$ is transformed into distinct low-rank representations using learnable projection matrices $\theta_K$, $\theta_V$, and $\theta_Q$, which govern the selective forgetting and remembering of input features. The resulting self-supervised loss is given by:

$$\ell(W; x_t) = \|f(\theta_K x_t; W) - \theta_V x_t\|^2, \ z_t = f(\theta_Q x_t; W_t). \qquad (13)$$

In the inner loop, only $W$ is optimized, so it is written as the parameter of $\ell$. And $\theta$ is the hyperparameter of this loss function. In the outer loop, $\theta_K$, $\theta_V$ and $\theta_Q$ are optimized along with other outer parameters, while $W$ is merely a hidden state. The TTT layer enhances the 1D-ResNet features by modeling their inter-dependencies, leading to a more enriched data representation that improves rPPG signal extraction. When reasoning on a target domain, the layer's hidden state weights adapt, allowing the model to learn the test set's real data distribution and significantly improve generalization and robustness for cross-domain applications. Finally, the layer outputs latent features of the original input dimension, from which the BVP waveform is estimated by the rPPG predictor.

## 4 EXPERIMENT

### 4.1 DATASET AND METRIC

The proposed method is evaluated on three standard datasets for rPPG-based heart rate estimation. **UBFC-rPPG** (Bobbia et al., 2019) includes 42 facial videos recorded at 30 fps (640×480). Participants were subjected to cognitive stress via a time-sensitive math game. **VIPL-HR** (Niu et al., 2020) contains 3,130 videos from 107 subjects, captured with multiple sensors under diverse conditions. For our experiments, we utilized the subsets recorded by the smartphone and depth camera. These videos are classified into three illumination levels: Lab (ceiling light on), Bright (all lights on), and Dim (all lights off). **MMPD** (Tang et al., 2023) includes 33 subjects with Fitzpatrick skin types ranging from 3 to 6, under four different lighting conditions and four different activities. It records 660 one-minute videos by using a mobile phone. **Evaluation Metrics.** The heart rate estimation performance is assessed using Mean Absolute Error (MAE), Root Mean Square Error (RMSE), Mean Absolute Percentage Error (MAPE), Pearson's correlation coefficient ($\rho$), and Signal-to-Noise Ratio (SNR). More implementation details are provided in Appendix A.1.

Table 1: Intra-dataset evaluation on the UBFC-rPPG, VIPL-HR and MMPD. Best results are marked in **bold** and second best in underline.

| Method | UBFC-rPPG | | | VIPL-HR | | | MMPD | | |
|---|---|---|---|---|---|---|---|---|---|
| | MAE↓ | RMSE↓ | $\rho\uparrow$ | MAE↓ | RMSE↓ | $\rho\uparrow$ | MAE↓ | RMSE↓ | $\rho\uparrow$ |
| CHROM | 8.20 | 9.92 | 0.27 | 11.40 | 16.90 | 0.28 | 13.66 | 18.76 | 0.08 |
| POS | 8.35 | 10.00 | 0.24 | 11.50 | 17.20 | 0.30 | 12.36 | 17.71 | 0.18 |
| DeepPhys | 6.27 | 10.82 | 0.65 | 11.00 | 13.80 | 0.11 | 22.27 | 28.92 | -0.03 |
| PhysNet | 2.95 | 3.67 | 0.97 | 10.80 | 14.80 | 0.20 | 4.80 | 11.80 | 0.60 |
| TS-CAN | 1.70 | 2.72 | 0.99 | - | - | - | 9.71 | 17.22 | 0.44 |
| PhysFormer | 0.50 | 0.71 | 0.99 | 4.97 | 7.79 | 0.78 | 11.99 | 18.41 | 0.18 |
| Dual-GAN | **0.44** | 0.67 | 0.99 | 4.93 | 7.68 | 0.81 | - | - | - |
| EfficientPhys | 1.14 | 1.81 | 0.99 | - | - | - | 13.47 | 21.32 | 0.21 |
| NEST | - | - | - | 4.76 | 7.51 | 0.84 | - | - | - |
| (Li & Yin, 2023) | 0.48 | **0.64** | 0.99 | 5.19 | 8.26 | 0.78 | - | - | - |
| RhythmMamba | 0.50 | 0.75 | 0.99 | 4.30 | 7.49 | 0.81 | **3.16** | **7.27** | **0.84** |
| **PhysTTT (Ours)** | 0.49 | 0.74 | 0.99 | **2.57** | **4.08** | **0.90** | 4.51 | 9.67 | 0.73 |

## 4.2 IN-DOMAIN EVALUATION

We conducted in-domain evaluation on the UBFC-rPPG, VIPL-HR and MMPD datasets to validate the feasibility of the proposed method PhysTTT. As shown in Table 1, for the UBFC-rPPG dataset, our method performs well in MAE, RMSE and $\rho$. The UBFC-rPPG dataset has clean facial videos as well as less noise, resulting in near saturation of the performance of the current state-of-the-art methods. To further evaluate the performance of PhysTTT, we employed two more challenging dataset. In the comparison with the previous methods, our method achieves the lowest MAE (2.57), RMSE (4.08), $\rho$ (0.90) and the second-lowest MAE (4.51), RMSE (9.67), $\rho$ (0.73) respectively on the VIPL-HR dataset and the MMPD dataset. In all three illumination scenarios, our method achieves performance beyond previous methods with MAEs of 1.94, 1.94, and 3.84 in the in-domain scenarios of Lab, Bright, and Dim, respectively. (More results and discussions can be found in Appendix A.2). This demonstrates that PhysTTT can accurately extract the weak rPPG signal features and understand their periodicity, which provides sufficient empirical evidence for the feasibility of the TTT-based architecture in the rPPG task.

## 4.3 CROSS-DOMAIN EVALUATION

Cross-domain evaluation is essential for assessing the generalization capability of PhysTTT, a prerequisite for its deployment in real-world applications where training and testing data may originate from different distributions. We set up four cross-dataset experiments. As shown in Table 2, in the case of UBFC-rPPG→VIPL-HR, PhysTTT consistently outperforms existing methods with the MAE (4.01), RMSE (7.85) and $\rho$ (0.67). In the case of VIPL-HR→UBFC-rPPG, PhysTTT achieves the MAE (0.98), RMSE (1.29) and $\rho$ (0.99). As shown in Table 3, we conducted the experiments by training on the UBFC-rPPG and VIPL-HR datasets, and evaluated on the MMPD dataset. In the case of UBFC-rPPG→MMPD, PhysTTT achieves the MAE (9.71), RMSE (16.70) and $\rho$ (0.36). In the case of VIPL-HR→MMPD, PhysTTT achieves the lowest MAE (10.35), RMSE (17.01) and $\rho$ (0.34). It can be seen that the results are much lower than other cases when tested on the MMPD dataset, since the noise and subjects in the MMPD datasets are much more complex and diverse. More evaluations and visualized results can be found in Appendix A.2 and Appendix B.

In summary, these cross-domain results indicate that PhysTTT effectively captures essential, domain-invariant physiological features, enabling reliable adaptation from simpler laboratory settings to more complex, noisy environments, and vice versa. Despite significant domain changes, PhysTTT maintains high accuracy comparable to in-domain performance. Cross-domain experiments demonstrate that PhysTTT has the ability to be deployed in real-world environments, which is particularly valuable for application scenarios such as remote health monitoring.

Table 2: Cross-dataset evaluation on UBFC-rPPG→VIPL-HR and VIPL-HR→UBFC-rPPG.

| Method | UBFC-rPPG→VIPL-HR | | | VIPL-HR→UBFC-rPPG | | |
|---|---|---|---|---|---|---|
| | MAE↓ | RMSE↓ | $\rho$↑ | MAE↓ | RMSE↓ | $\rho$↑ |
| DeepPhys | 15.09 | 19.03 | 0.05 | 15.09 | 18.87 | 0.31 |
| PhysNet | 14.54 | 22.75 | -0.04 | 17.70 | 20.68 | 0.29 |
| TS-CAN | 11.95 | 19.23 | 0.16 | 19.25 | 21.67 | -0.02 |
| PhysFormer | 12.00 | 15.35 | 0.10 | 22.62 | 25.84 | 0.06 |
| EfficientPhys | 6.43 | 11.86 | 0.58 | 15.02 | 20.54 | 0.36 |
| NEST | 10.60 | 13.60 | 0.37 | 7.45 | 9.51 | 0.75 |
| PhysMamba | 18.38 | 23.02 | 0.02 | 12.71 | 14.90 | 0.05 |
| Bi-TTA | 8.15 | 12.82 | 0.50 | 8.05 | 18.53 | 0.77 |
| TTA-rPPG | - | - | - | 5.60 | 7.16 | - |
| RhythmMamba | 5.71 | 11.36 | 0.55 | 0.90 | 1.51 | 0.99 |
| PhysTTT (Ours) | 4.01 | 7.85 | 0.67 | 0.98 | 1.29 | 0.99 |

Table 3: Cross-dataset evaluation on UBFC-rPPG→MMPD and VIPL-HR→MMPD.

| Method | UBFC-rPPG→MMPD | | | VIPL-HR→MMPD | | |
|---|---|---|---|---|---|---|
| | MAE↓ | RMSE↓ | $\rho$↑ | MAE↓ | RMSE↓ | $\rho$↑ |
| DeepPhys | 17.50 | 25.00 | 0.05 | 16.23 | 19.86 | 0.13 |
| PhysNet | 9.47 | 16.01 | 0.31 | 15.30 | 19.77 | -0.05 |
| TS-CAN | 14.01 | 21.04 | 0.24 | 15.68 | 20.01 | 0.00 |
| PhysFormer | 12.10 | 17.79 | 0.17 | 19.62 | 23.14 | 0.06 |
| EfficientPhys | 13.78 | 22.25 | 0.09 | 17.88 | 23.29 | 0.02 |
| PhysMamba | 11.96 | 17.69 | 0.29 | 15.03 | 18.08 | -0.17 |
| SpikingPhys | 14.15 | 16.22 | 0.15 | - | - | - |
| RhythmMamba | 10.63 | 17.14 | 0.34 | 10.87 | 17.57 | 0.33 |
| PhysTTT (Ours) | 9.71 | 16.70 | 0.36 | 10.35 | 17.01 | 0.34 |

## 4.4 ABLATION STUDY

**Impact of Key Modules.** We conduct an ablation experiment taking the Lab domain in the VIPL-HR dataset as an example to explore the impact of different modules in the model on the measurement accuracy. As shown in Table 4, each module has played a certain role. When frame stem is not used for preprocessing, that is, simply using convolution operations on the input video frames to align the output feature dimension with the input dimension of the backbone network, the performance of the model is the worst. When extracting inter-frame differences as the input feature, the performance is improved. In the replacement of the backbone network, the performance is improved when using the 1D-ResNet designed in this paper compared with using a single-layer 1D-CNN and not using any backbone network. In the comparison with different TTA variants, we use Tent or SAR methods to adapt the pre-trained models containing fusion stem and 1D-ResNet to the target domain, and the results are significantly worse than those of the models containing TTT layers. Frame stem focuses on the ROI region of the human face and extracts fine-grained inter-frame difference features, magnifying the minute changes in skin color. 1D-ResNet focuses on the rPPG representation in skin pixels and further extracts deep-level information. The TTT layer can better capture the dependencies between each frame, enhance the interaction between features, facilitate the extraction of periodic patterns of BVP signals, and through training during testing, improve the generalization ability of the model.

**Impact of Noise.** We explore the impact of different lights, motions, and skin tones, In the test, we use the model trained on the UBFC-rPPG dataset. When testing a certain type, conditions in any other type are not excluded. Table 5 shows the results in different tasks. The results show that PhysTTT can generalize well in indoor lighting environments, stationary, small head movements and lighter skin tone, but performs not well in some challenging scenarios (such as natural light and

Table 4: Impact of key modules.

| Frame Stem | Backbone | Method | MAE↓ | RMSE↓ | MAPE↓ | $\rho$↑ | SNR↑ |
|---|---|---|---|---|---|---|---|
| × | 1D-ResNet | TTT | 14.12 | 17.67 | 20.86 | 0.12 | -11.51 |
| Differential Stem | 1D-ResNet | TTT | 10.44 | 14.66 | 15.69 | 0.04 | -10.78 |
| Fusion Stem | × | TTT | 3.38 | 6.13 | 4.82 | 0.82 | -1.79 |
| Fusion Stem | 1D-CNN | TTT | 2.61 | 4.60 | 3.65 | 0.87 | 1.06 |
| Fusion Stem | 1D-ResNet | × | 8.58 | 16.63 | 12.86 | 0.25 | -5.12 |
| Fusion Stem | 1D-ResNet | Tent | 8.67 | 16.62 | 12.97 | 0.26 | -4.65 |
| Fusion Stem | 1D-ResNet | SAR | 7.70 | 15.32 | 11.62 | 0.28 | -4.65 |
| Fusion Stem | 1D-ResNet | TTT | **1.94** | **3.64** | **2.82** | **0.93** | **1.32** |

Table 5: Impact of different lights, motions and skin tones.

| Noise | Type | MAE↓ | RMSE↓ | MAPE↓ | $\rho$↑ | SNR↑ |
|---|---|---|---|---|---|---|
| Light | LED-low | 9.29 | 15.3 | 10.4 | 0.38 | -7.66 |
| | LED-high | 9.24 | 15.29 | 9.88 | 0.54 | -6.85 |
| | Incandescent | 7.68 | 14.72 | 8.11 | 0.39 | -5.59 |
| | Nature | 12.65 | 20.75 | 12.59 | 0.19 | -7.87 |
| Motion | Stationary | 5.85 | 12.83 | 5.80 | 0.59 | -3.95 |
| | Rotation | 6.17 | 10.44 | 7.25 | 0.57 | -6.14 |
| | Talking | 5.84 | 11.52 | 6.52 | 0.43 | -3.81 |
| | Walking | 21.60 | 25.51 | 22.68 | 0.00 | -15.07 |
| Skin tone | 3 | 5.33 | 11.90 | 6.01 | 0.59 | -2.61 |
| | 4 | 12.86 | 19.06 | 13.40 | 0.21 | -10.50 |
| | 5 | 13.01 | 19.86 | 16.30 | 0.26 | -9.42 |
| | 6 | 14.18 | 21.11 | 14.36 | 0.25 | -11.83 |

walking). Since the training data (UBFC-rPPG) only included subjects with skin types 2-3 and most of them are stationary, which affect the generalization ability of the model to a certain extent. More evaluations and discussions can be found in Appendix A.3 and Appendix A.4.

## 4.5 COMPUTATIONAL COST

We perform a 10-second inference test using a sequence of image frames from $X \in \mathbb{R}^{300 \times 3 \times 128 \times 128}$. Table 6 shows the number of model parameters, average multiply-accumulate operations (MACs) per frame, average throughput per frame, average peak GPU memory usage per frame and average delay per frame of different methods. PhysTTT is at the top level in terms of average MACs per frame (42.64M) and average peak GPU memory usage per frame (7.08M). Although its throughput and delay are relatively lower than other methods, but it should be noted that the PhysTTT strikes a balance between measurement accuracy and inference speed, increasing some of the inference time for training at the time of testing, but improving substantially in accuracy. Moreover, a delay of 0.24ms per frame is sufficient to meet the requirements of actual deployment, demonstrating the potential for reliable application in real-world scenarios.

## 5 CONCLUSION

In response to the difficulty of existing rPPG methods to balance measurement accuracy and computational cost in cross-domain scenarios, we propose PhysTTT, an accurate heart rate measurement framework adapted to multi-scenarios. The method combines multiple 1D-CNNs containing residual structures and TTT layer, and improves cross-domain generalization by learning the real data distribution at test time that is unseen at the training time. In-domain and cross-domain experiments show that PhysTTT achieves excellent performance levels with high accuracy and good robustness. In the future, we will further explore its applicability in real-world environments and feasibility of deployment on mobile devices.

## ETHICS STATEMENT

All the data used in this study are sourced from the public dataset (UBFC-rPPG (Bobbia et al., 2019), VIPL-HR (Niu et al., 2020) and MMPD (Tang et al., 2023)), which have undergone appropriate ethical review and anonymization processing. We adhere to data privacy regulations and promote transparency through open-source releases. We strongly advocate responsible use solely for beneficial healthcare applications. Our study followed established research ethics guidelines, and we declare no conflicts of interest. We encourage on going interdisciplinary dialogue to address potential risks and ensure responsible development and deployment of such technologies, recognizing the broader societal impacts of AI in healthcare. We remain committed to ethical AI advancement and welcome further discussion on the critical issues, including the development of governance frameworks to prevent misuse and protect data privacy.

## REPRODUCIBILITY STATEMENT

Models, data, and code are publicly available for reproducibility and future research. We exclusively utilize publicly accessible datasets, which can be requested or downloaded from the respective study group websites, allowing others to easily obtain the data for their own analyses. In the Section 4 and Appendix A.1, we provide comprehensive descriptions of the datasets, implementation details, and data pre-processing methods used in our experiments, ensuring transparency in our data handling procedures. The code to run our model is published with user-friendly examples. We have provided a detailed overview of the model architecture and its hyperparameters in the Section 3, Thus, our work is designed to be reproducible, enabling future research to build upon our findings.

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

# A APPENDIX

## A.1 IMPLEMENTATION DETAILS

All experiments are implemented based on the open-source deep physiological sensing toolbox, rPPG-Toolbox (Liu et al., 2023b). We adhere to the evaluation protocol established by Lu et al. (2021). For the UBFC-rPPG dataset, the first 30 subjects were used for training, and the remaining 12 were held out for testing. As no dedicated validation set is available, the model checkpoint from the final training epoch is selected for evaluation. For the VIPL-HR dataset, 86 subjects are used for training and 21 for testing in each scenario. Following the subject-specific 5-fold cross-validation protocol (Niu et al., 2020), and considering computational constraints, we report results from fold-1. For the MMPD dataset, a total of 460 videos were used for training, 60 for validation, and the remaining 140 for testing (Zou et al., 2025a). Due to computational constraints, we utilized the mini-MMPD version with a resolution of 320×240 and a frame rate of 30Hz for experimentation. All videos are segmented into 10-second clips for analysis. Heart rate is estimated by first applying a bandpass filter (0.75–2.5 Hz, corresponding to 45–150 BPM) to the raw signals, then applying the Fast Fourier Transform (FFT), with the result measured in BPM. The model is trained using the AdamW optimizer with a learning rate of $5 \times 10^{-5}$ for 30 epochs, the incorporated weight decay in AdamW helps mitigate overfitting by adding L2 regularization. The training process is conducted on NVIDIA GeForce RTX 3090 GPU with a batch size of 4.

Four cross-domain cases were set up in the experiments. Case-1 uses the first 30 subjects in the UBFC-rPPG dataset as the training set, and the three illumination scenarios (Lab, Bright, and Dim) of fold-1 in the VIPL-HR dataset as the testing set (UBFC-rPPG→VIPL-HR). Case-2 uses three illumination scenes from 86 subjects in the VIPL-HR dataset as the training set, and the UBFC-rPPG dataset as the test set (VIPL-HR→UBFC-rPPG). The training set settings for cases-3 (UBFC-rPPG→MMPD) and case-4 (VIPL-HR→MMPD) follow those of case-1 and case-2, but the test sets for both are the MMPD dataset.

## A.2 DIFFERENT ILLUMINATION EVALUATION

We evaluated the performance of PhysTTT in different illumination scenarios in the VIPL-HR dataset. As shown in Table 7, PhysTTT maintains good performance and stability in both intra-illumination and cross-illumination experiments. In the case of intra-illumination, our method achieves the lowest MAEs of 1.94, 1.94, and 3.84 in the Lab, Bright, and Dim scenarios respectively. In the case of cross-illumination, PhysTTT still performs well compared with the previous method. For instance, when trained on both Bright and Dim scenarios and tested on the Lab scenario, PhysTTT achieves the lowest MAEs of 2.75 and 2.97, respectively. These values represent only a minor performance degradation of 0.81 and 1.03 compared to its performance when trained and tested within the Lab domain. Similarly, in the other cross-illumination cases, the model maintains the lowest MAEs (e.g., 2.65 and 2.57 for Bright; 3.36 and 3.68 for Dim). It corroborates that PhysTTT possesses not only high prediction accuracy but also exceptional stability across diverse illumination conditions.

## A.3 IMPACT OF LOSS FUNCTIONS

We compare the impact of different loss functions on the MMPD dataset. Table 8 shows that the performance degradation is more severe when using only $\mathcal{L}_{peak}$ than when using only $\mathcal{L}_{np}$ or $\mathcal{L}_{freq}$. However, after aligning the signals from coarse-grained perspectives such as trend and frequency, $\mathcal{L}_{peak}$ can further fit the waveforms at the fine-grained level, thereby improving the accuracy of recognition.

## A.4 IMPACT OF CHUNK LENGTHS.

We consider the impact of the input length of the model. We test the trained PhysTTT model on videos with different chunk lengths. As shown in Table 9, the trained model can adapt to video clips of most lengths without reducing performance. Some performance degradation will only occur when the length increases to more than 20 seconds. The results of different chunk lengths clearly indicate that PhysTTT can effectively learn the quasi-periodic patterns of rPPG.

Table 6: Computational cost on Parameters, MACs, Throughput, Peak GPU memory usage and Delay.

| Method | Param.(M) | MACs(M) | Throughput(Kfps) | Memory(M) | Delay(ms) |
|---|---|---|---|---|---|
| DeepPhys | 1.98 | 762.32 | 6.29 | 35.64 | 0.16 |
| PhysNet | 0.75 | 448.76 | **14.4** | 11.74 | **0.07** |
| TS-CAN | 1.98 | 762.32 | 5.83 | 38.56 | 0.17 |
| PhysFormer | 7.38 | 323.88 | 9.63 | 22.32 | 0.10 |
| EfficientPhys | 1.91 | 382.69 | 9.73 | 25.60 | 0.10 |
| PhysMambda | **0.57** | 323.35 | 8.08 | 12.02 | 0.12 |
| RhythmMambda | 1.07 | 82.84 | 7.56 | 7.66 | 0.13 |
| **PhysTTT (Ours)** | 1.92 | **42.64** | 4.23 | **7.08** | 0.24 |

Table 7: In-domain and cross-domain evaluation under three illumination scenarios on the VIPL-HR. (In-domain: Lab→Lab, Bright→Bright, Dim→Dim; Cross-domain: Lab→Bright, Lab→Dim, Bright→Lab, Bright→Dim, Dim→Bright, Dim→Lab).

| Method | Domain | Train | | | | | | | | |
|---|---|---|---|---|---|---|---|---|---|---|
| | | Lab | | | Bright | | | Dim | | |
| | Test | MAE↓ | RMSE↓ | ρ ↑ | MAE↓ | RMSE↓ | ρ ↑ | MAE↓ | RMSE↓ | ρ ↑ |
| DeepPhys | Lab | 16.82 | 21.80 | 0.06 | 18.81 | 21.72 | 0.06 | 17.08 | 21.97 | -0.10 |
| | Bright | 15.67 | 19.70 | 0.06 | 21.94 | 25.46 | -0.35 | 19.87 | 24.05 | 0.20 |
| | Dim | 22.37 | 28.45 | 0.11 | 18.27 | 23.24 | -0.02 | 19.03 | 23.91 | 0.02 |
| PhysNet | Lab | 9.31 | 11.29 | 0.47 | 10.44 | 13.71 | -0.08 | 11.53 | 14.42 | 0.01 |
| | Bright | 13.64 | 17.38 | 0.05 | 9.63 | 12.63 | 0.08 | 13.78 | 17.33 | 0.00 |
| | Dim | 11.03 | 14.28 | 0.25 | 9.82 | 13.12 | 0.23 | 11.02 | 13.12 | 0.33 |
| PhysFormer | Lab | 10.65 | 14.76 | -0.01 | 11.07 | 13.77 | -0.21 | 13.18 | 15.92 | -0.20 |
| | Bright | 11.46 | 14.78 | 0.11 | 13.14 | 16.01 | -0.30 | 13.13 | 17.04 | -0.04 |
| | Dim | 12.13 | 15.94 | -0.29 | 10.42 | 13.44 | -0.08 | 12.52 | 15.43 | 0.27 |
| EfficientPhys | Lab | 7.12 | 13.27 | 0.44 | 14.58 | 19.41 | -0.15 | 12.87 | 17.13 | 0.07 |
| | Bright | 10.89 | 17.27 | 0.10 | 10.87 | 14.22 | 0.45 | 11.71 | 15.72 | 0.42 |
| | Dim | 11.51 | 16.96 | 0.36 | 16.57 | 20.90 | -0.10 | 10.36 | 14.54 | 0.00 |
| RhythmMamba | Lab | 16.45 | 21.84 | 0.01 | 12.31 | 15.70 | 0.11 | 14.66 | 19.21 | 0.10 |
| | Bright | 18.76 | 22.82 | -0.19 | 14.88 | 19.50 | 0.21 | 19.40 | 26.88 | 0.00 |
| | Dim | 23.31 | 28.04 | -0.15 | 18.36 | 22.13 | 0.22 | 11.28 | 16.74 | 0.37 |
| **PhysTTT (Ours)** | Lab | **1.94** | **3.64** | **0.93** | **2.75** | **4.54** | **0.87** | **2.97** | **6.35** | **0.85** |
| | Bright | **2.65** | **3.55** | **0.95** | **1.94** | **2.54** | **0.97** | **2.57** | **3.66** | **0.94** |
| | Dim | **3.36** | **4.73** | **0.92** | **3.68** | **5.96** | **0.80** | **3.84** | **6.06** | **0.81** |

Table 8: Impact of different loss functions.

| $\mathcal{L}_{np}$ | $\mathcal{L}_{freq}$ | $\mathcal{L}_{peak}$ | MAE↓ | RMSE↓ | MAPE↓ | $\rho$ ↑ | SNR↑ |
|---|---|---|---|---|---|---|---|
| ✓ | × | × | 6.30 | 13.05 | 6.73 | 0.59 | -3.77 |
| × | ✓ | × | 7.54 | 14.01 | 7.70 | 0.45 | -4.23 |
| × | × | ✓ | 18.08 | 22.57 | 21.23 | -0.09 | -13.24 |
| ✓ | ✓ | × | 5.84 | 11.82 | 6.14 | 0.61 | -1.42 |
| ✓ | ✓ | ✓ | **4.51** | **9.67** | **5.07** | **0.73** | **-0.57** |

Table 9: Impact of different chunk lengths.

| Chunk length | MAE↓ | RMSE↓ | MAPE↓ | $\rho$ ↑ | SNR↑ |
|---|---|---|---|---|---|
| 60 (2s) | 0.59 | 0.87 | 0.59 | 0.99 | 6.57 |
| 120 (4s) | 0.50 | 0.71 | 0.51 | 0.99 | 6.32 |
| 180 (6s) | 0.59 | 0.81 | 0.61 | 0.99 | 6.58 |
| 240 (8s) | 0.59 | 0.81 | 0.59 | 0.99 | 6.64 |
| 300 (10s) | 0.49 | 0.74 | 0.50 | 0.99 | 7.34 |
| 600 (20s) | 0.76 | 0.97 | 0.78 | 0.99 | 6.40 |
| 900 (30s) | 1.13 | 1.90 | 1.21 | 0.98 | 3.95 |

## B  VISUALIZED RESULTS

To further analyze the performance of PhysTTT, we visualize the in-domain prediction results on the UBFC-rPPG dataset, adopting a second-order Butterworth filter (cut-off frequency: 0.75 and 2.5 Hz). The Figure 6 shows the waveforms and power spectral densities of the predicted BVP signals and the real BVP signals, and the Pearson correlation coefficient between them can reach 0.76. As shown in Figure 4 and Figure 5, the Bland-Altman difference plots and scatter plots in various cases show a strong correlation between the predicted heart rate and the actual heart rate. These results confirm the accuracy and reliability of PhysTTT in heart rate measurement.

## C  THE USE OF LARGE LANGUAGE MODELS (LLMs)

The authors employed a large language model (DeepSeek-V3.1-Terminus) exclusively for the purpose of text polishing and refinement after the intellectual content of the paper was fully developed by the human authors.

The LLM was used to:

- Improve the clarity, fluency, and grammatical correctness of the manuscript.
- Suggest alternative phrasing for awkward or repetitive sentences.
- Ensure consistency in academic tone and style.

The LLM did not contribute to the following, which were solely the work of the authors:

- Formulation of the research ideas, hypotheses, or methodology.
- Conducting the literature review, data analysis, or interpretation of results.
- Creation of figures, tables, or any original scientific content.

All final editorial decisions were made by the authors. The LLM served strictly as a writing assistance tool, and the intellectual substance of the work remains entirely attributable to the human authors.

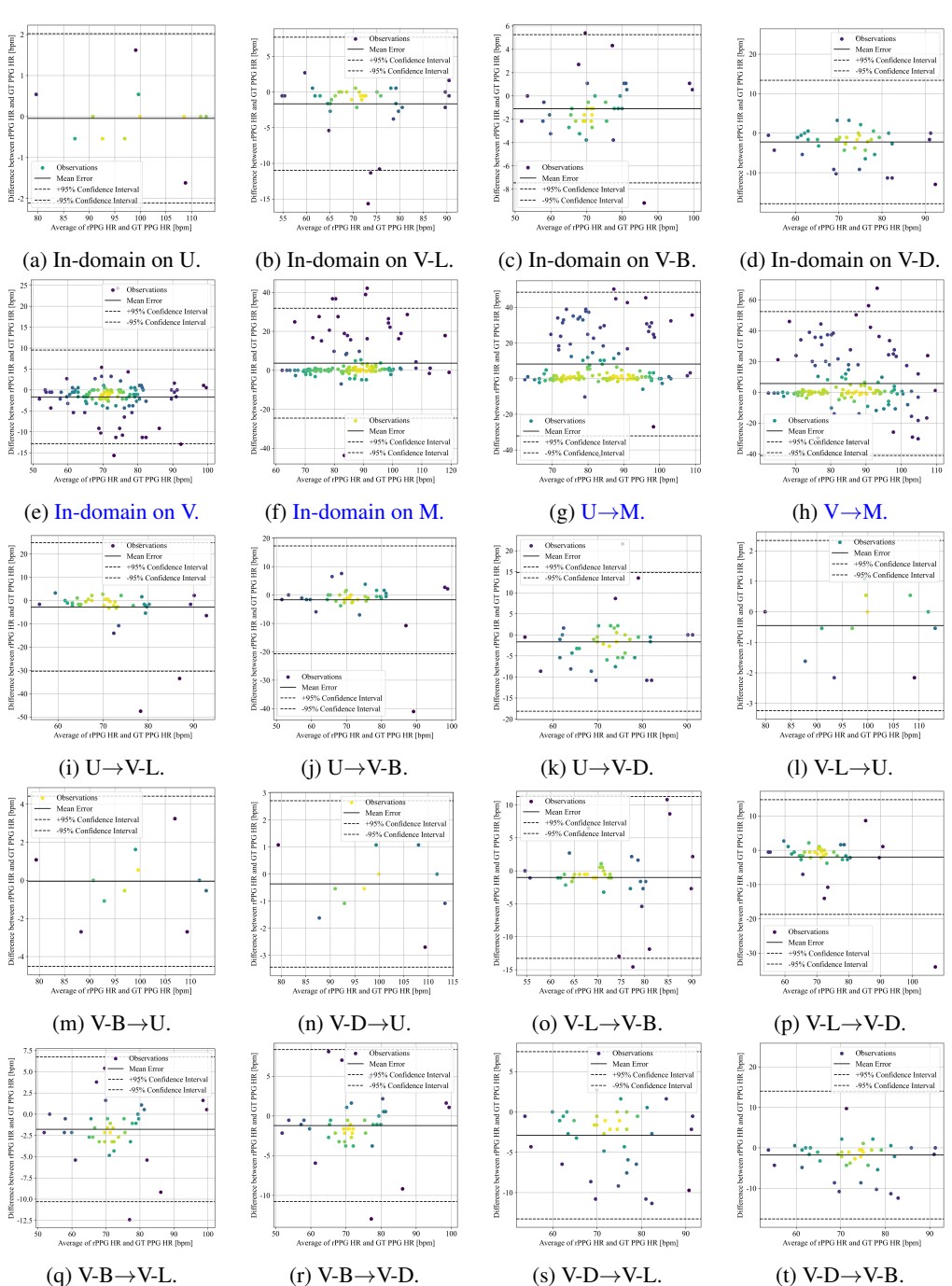

Figure 4: Bland-Altman plots of results. (UBFC-rPPG: U, VIPL-HR: V, VIPL-Lab: L, VIPL-Bright: B, VIPL-Dim: D, MMPD: M)

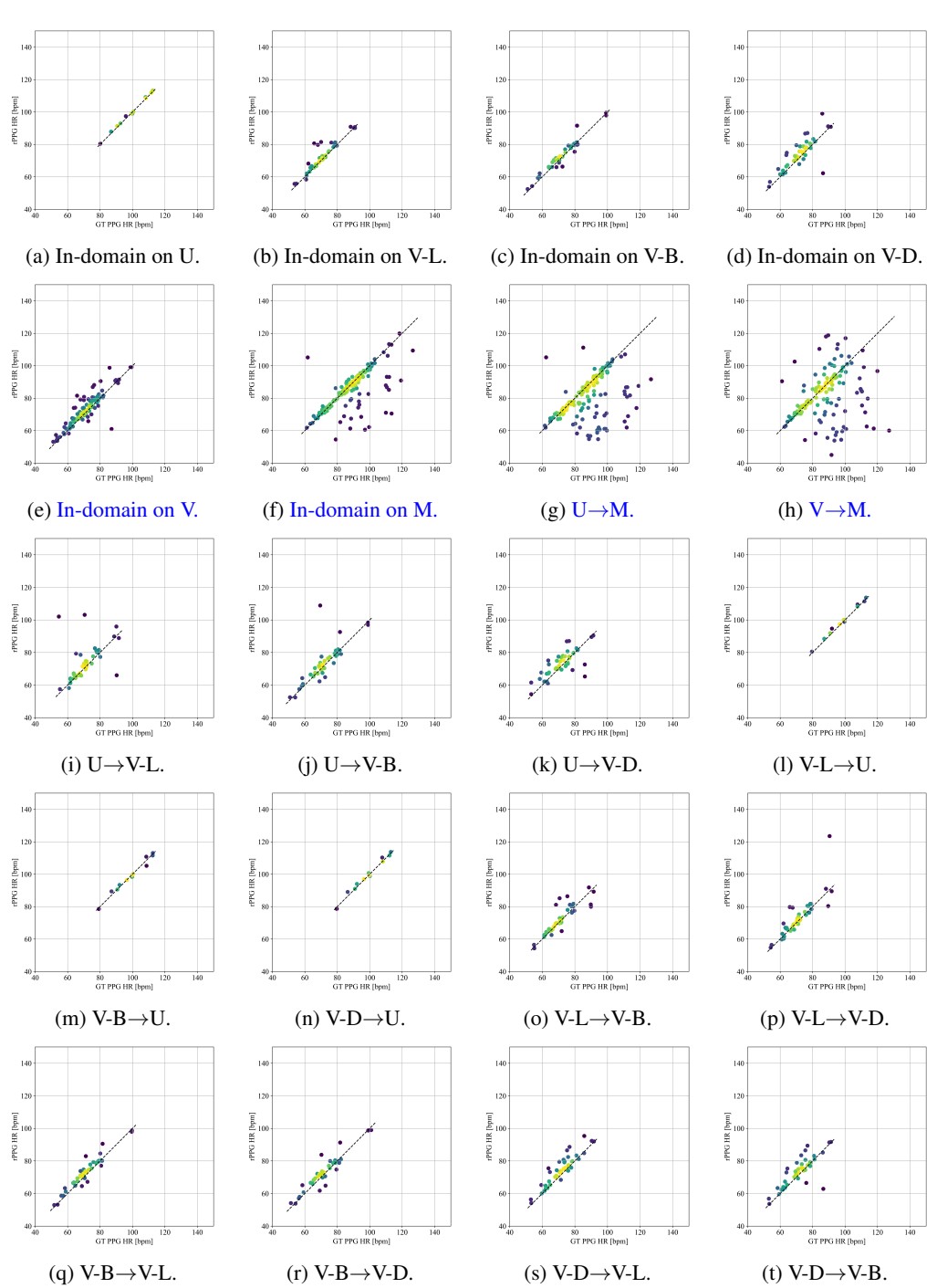

Figure 5: Scatter plots of results. (UBFC-rPPG: U, VIPL-HR: V, VIPL-Lab: L, VIPL-Bright: B, VIPL-Dim: D, MMPD: M)

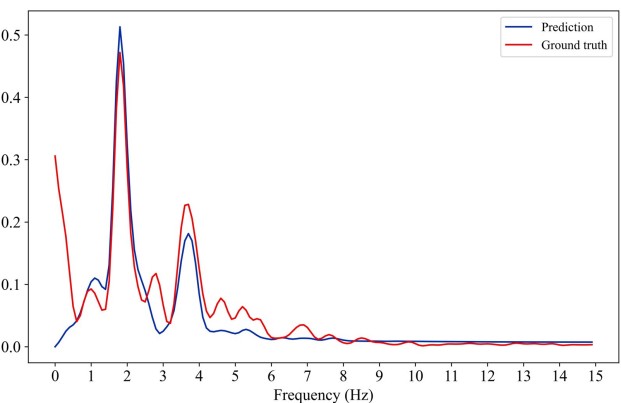

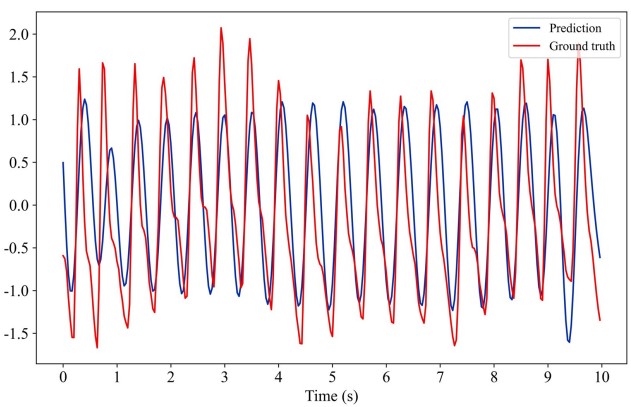

Figure 6: An example of results on UBFC-rPPG

