# OpenReview forum: "PhysTTT: Accurate and Lightweight Cross-Domain Heart Rate Measurement with Test-Time Training"
_ICLR.cc/2026/Conference — Submitted to ICLR 2026_

### Official Review · Reviewer_pzPR · 2025-10-29

**Soundness:** 3
**Presentation:** 3
**Contribution:** 3
**Rating:** 6
**Confidence:** 4

**Summary:**

This work introduces a lightweight rPPG measurement framework that leverages test-time training to enhance adaptability in cross-domain scenarios. The method focuses on enforcing temporal consistency through multi-objective losses that align waveform trends and spectral components with physiological patterns. The approach shows promise for practical deployment due to its low complexity and ability to update online at inference time (Table 5).

**Strengths:**

1. The model achieves competitive performance with a compact architecture, demonstrating a strong balance between computational efficiency and prediction accuracy suitable for real-time or mobile deployment.

2. By updating parameters during inference, the method adapts dynamically to unseen conditions and distributions, improving robustness when deployment environments differ from the training domain.

**Weaknesses:**

1. The frequency and waveform stability is mainly enforced by the definition of the lost function. While the model itself have simple architectures without explicit ability to help on the temporal structure of rPPG. A better design or an explanation of why keeping the model architecture simple is needed.

2. The cross-domain evaluation currently focuses only on UBFC-rPPG and VIPL-HR. These datasets share limited diversity in terms of illumination, subject motion, and demographic variations. As a result, the reported cross-domain gains may largely reflect spectral alignment improvements rather than true robustness to heterogeneous real-world conditions. Including more challenging and diverse datasets such as MMPD or BUAA would provide a more comprehensive assessment of generalization performance, particularly regarding sensitivity to skin tone variation, lighting shifts, and motion artifacts, which are key factors in practical deployment.

**Questions:**

1. What is the reason of separating the optimization process into an inner-outer loop? Does the optimization of W and theta be split to ensure convexity or smoothness? Or it is simply based on empirical observation?

**Details Of Ethics Concerns:**

No concerns.

---

> ### Author Response · Authors · 2025-11-22
> **Response to Reviewer pzPR (1/2)**
>
> Dear Reviewer pzPR,
>
> We are extremely grateful for your review of the paper. We hope our responses below address your concerns about our paper.
>
> > **W1: The frequency and waveform stability is mainly enforced by the definition of the lost function. While the model itself have simple architectures without explicit ability to help on the temporal structure of rPPG. A better design or an explanation of why keeping the model architecture simple is needed.**
>
> **A1:** To better estimate the characteristics of the BVP signal, we jointly optimize the supervised loss of the model by defining the loss function $L_{np}$ that maintains trend consistency, the $L_{freq}$ that maintains frequency consistency, and the $L_{peak}$ that maintains waveform consistency. Although the model itself has a simple architecture, it is important to note the introduction of the TTT layer.
>
> The essence of the TTT layer is a sequence modeling layer similar to RNN. Since all sequence modeling layers can store historical context in a hidden state, this also indicates the ability of the TTT layer in time modeling. Moreover, the TTT layer replaces the original hidden state in RNN with a machine learning model This structure, which eliminates the performance limitation of the RNN layer in long contexts due to its fixed-size hidden state expression, also expands its modeling ability for long time series coherence and periodic data.
>
> Due to the extremely weak rPPG signal, the signal-to-noise ratio is very low. Choosing a complex model structure may cause overfitting of various noises in the data (such as motion artifacts and lighting changes), rather than the essential physiological signals that we are most concerned about. Simple models, on the one hand, play the role of strong priors and filters, forcing the network to focus on the most significant features, which may be more robust in scenarios such as unseen lighting, skin color, and motion. On the other hand, they reduce the number of model parameters and computational costs, which is conducive to deployment on devices with limited computing resources.
>
> We add a description of the time series modeling capability of the TTT layer in the revised version. Please refer to Section 3.4 for details.
>
> > **W2: The cross-domain evaluation currently focuses only on UBFC-rPPG and VIPL-HR. These datasets share limited diversity in terms of illumination, subject motion, and demographic variations. As a result, the reported cross-domain gains may largely reflect spectral alignment improvements rather than true robustness to heterogeneous real-world conditions. Including more challenging and diverse datasets such as MMPD or BUAA would provide a more comprehensive assessment of generalization performance, particularly regarding sensitivity to skin tone variation, lighting shifts, and motion artifacts, which are key factors in practical deployment.**
>
> **A2:** The issues regarding datasets and diverse scenarios are common problems. We have added intra-dataset and cross-dataset experiments in the MMPD dataset and tested the robustness of the model in different complex scenarios. Please refer to the first point of our Common Response for details.

---

> > ### Author Response · Authors · 2025-11-22
> > **Response to Reviewer pzPR (2/2)**
> >
> > > **Q1: What is the reason of separating the optimization process into an inner-outer loop? Does the optimization of W and theta be split to ensure convexity or smoothness? Or it is simply based on empirical observation?**
> >
> > **A3:** **Separating the optimization process into inner and outer loops enables the model to continuously learn and adapt during testing.** This approach overcomes the limitation of insufficient hidden state expression ability of RNNs in long context sequences and avoids the secondary computational complexity problem existing in the self-attention mechanism.
> >
> > The TTT layer has the same interface as other sequence modeling layers such as the RNN layer and the self-attention mechanism, so it can be replaced in any larger network architecture. The way to train the network using the TTT layer is also the same as that for training other models. From the perspective of the overall network architecture, training a larger network is regarded as an outer loop, and training the model parameters $W$ within the TTT layer is regarded as an inner loop. **The outer loop aims to optimize the initial parameters of the network and learn a general hidden state update strategy, enabling the inner loop to effectively update the state through self-supervised tasks. The inner loop utilizes the strategies learned from the outer loop to optimize the hidden state during testing through self-supervised learning, thereby enhancing the adaptability to the current test sequence.**
> >
> > The inner loop gradient $\nabla \ell$ is about the parameter $W$ of each machine learning model $f$ in the TTT layer, while the outer loop gradient is about the parameter $\theta_{rest}$ of the rest of the network. The goal of TTT is to ensure that $z_t=f(x_t; W_t)$ performs well in time series modeling. Therefore, optimizing $W$ becomes the most important self-supervised task in the TTT layer, as it determines the type of features that $W$ will learn from the test sequence. One end-to-end approach is to directly optimize the self-supervised task to achieve the prediction of $z_t$. Specifically, we learn the self-supervised task as part of the outer loop, with the parameters $\theta_{K}, \theta_{V}, \theta_{Q}$ of the loss $\ell(W; x_t)$ as the parameters of the outer loop and $W$ as the parameter of the inner loop. During training, only $W$ in the inner loop is optimized, so it is written as the parameter of $\ell$. $\theta$ is the hyperparameter of this loss function. In the outer loop, $\theta_{K}, \theta_{V}, \theta_{Q}$ and $\theta_{rest}$ are optimized together, and $W$ is merely a hidden state rather than a parameter.
> >
> > **This split of $\theta$ and $W$ optimization is mainly based on empirical observation and inductive bias in architecture, but they have implicitly had a positive impact on convexity and smoothness in practice.** $\theta$ learns the prior knowledge that is universal across sequences and how to adapt to new sequences, while $W$ learns the temporary and specific knowledge for the current test sequence. Although convexity and smoothness were not the original intentions of the split, this design influenced the optimization process. In the TTT-MLP in this paper, although the inner loop problem is not convex, it is usually much simpler than the outer loop problem (with fewer parameters and more specific tasks). This relative simplicity enables the internal circulation optimization to remain relatively stable and rapid, thereby indirectly assisting the external circulation. The loss function $\mathcal{L}(\theta)$ of the outer loop is not a direct fit to the original data. Instead, it specifies the information in $x_t$ through training view $\theta_Kx_t$ and label view $\theta_Vx_t$, which is compressed into $W_t$ and propagated forward over time. The test view $\theta_Qx_t$ specifies the potentially different information mapped to the current output $z_t$ and propagated forward through the network layer, thereby affecting the final loss. This process may play a smoothing role, making $\mathcal{L}(\theta)$ smoother than directly fitting all the data, and adding more flexibility to self-supervised tasks.
> >
> > We have supplemented the role of the inner and outer loop in the revised edition. Please refer to Section 3.4 for details.
> >
> > Thank you again for your valuable feedback and insights.

---

### Official Review · Reviewer_jTxE · 2025-10-31

**Soundness:** 3
**Presentation:** 2
**Contribution:** 2
**Rating:** 4
**Confidence:** 5

**Summary:**

This paper introduces a novel Test-Time Training (TTT) framework for remote physiological signal measurement. The proposed approach employs multi-frame difference fusion techniques and leverages 1D-CNNs to extract spatio-temporal features from facial video sequences. The TTT layer subsequently compresses contextual information from video frame sequences into a self-supervised learning model, enabling adaptation to unseen domain distributions during the testing phase. Experimental results demonstrate that the method outperforms existing state-of-the-art approaches, exhibiting precise and efficient measurement capabilities.

**Strengths:**

1. The method systematically introduces Test-Time Training into the Test-Time Adaptation framework for remote photoplethysmography (rPPG), effectively addressing domain shift issues in this field.
2. The proposed approach achieves a favorable balance between measurement accuracy and computational cost, thereby enhancing the generalizability of remote physiological signal measurement, which is particularly valuable for real-world deployment.
3. Comprehensive experiments validate the exceptional performance of the method in both in-domain and cross-domain scenarios.
4. The paper is well-written with a clear narrative structure, adequately covers related work, and distinctly highlights its contributions.

**Weaknesses:**

1. In lines 084–086 of the introduction, the authors state that “While SSMs offer a better balance between linear complexity and long-range modeling, they can face limitations in generalization and parallel processing, particularly in cross-domain scenarios where model robustness is critical.” This claim lacks sufficient justification or empirical evidence.
2. In line 088 of the introduction, there are several citation errors. Specifically, Dual-GAN is a standard supervised learning method, while Dual-bridging belongs to the domain generalization category, not domain adaptation, as stated by the authors.
3. Section 2.2 of the related work is unclear and lacks logical structure. It is not evident what the authors intend to convey, nor why “PhysTTT effectively captures spatial information.” This section needs a clearer focus and stronger justification.
4. The Frame Stem module described in Section 3.2 is nearly identical to those used in RhythmFormer and RhythmMamba. This component should not be claimed as a novel contribution, and the authors must explicitly clarify this overlap in the paper.
5. All formulas in the paper are missing serial numbering, which violates the standard formatting and submission requirements of academic papers.
6. Both the intra-dataset and cross-dataset experiments only consider illumination variations, without evaluating robustness under other common noise factors such as head motion. This is a significant limitation in the experimental design.
7. The paper reports cross-domain experiments only on UBFC-rPPG and VIPL-HR, which are not sufficiently diverse. It is recommended to validate the proposed method on a broader range of datasets (e.g., MMPD, MR-NIRP-Car) to demonstrate robustness and generalization.
8. Table 5 mentions “Throughput” in its caption, but the corresponding quantitative results are missing.
9. The paper lacks crucial ablation studies on these core modules and the self-supervised learning strategy, as well as validation on alternative backbones to verify the generality of the proposed approach.

**Questions:**

1. The paper does not clearly explain the conceptual and practical differences between TTT (Test-Time Training) and TTA (Test-Time Adaptation). What are the respective advantages and disadvantages of TTT compared to TTA, and why is TTT considered more suitable for rPPG tasks?
2. The source of the model’s strong cross-domain performance remains unclear. To what extent does it stem from the carefully designed 1D-ResNet and frame-difference backbone, and to what extent from the TTT adaptation layer?

---

> ### Author Response · Authors · 2025-11-22
> **Response to Reviewer jTxE (1/4)**
>
> Dear Reviewer jTxE,
>
> We are extremely grateful for your review of the paper. You have raised a number of important issues. We agree with your comments and have modified our paper accordingly. Below we give a point-by-point response to your concerns and suggestions.
>
> > **W1: In lines 084–086 of the introduction, the authors state that “While SSMs offer a better balance between linear complexity and long-range modeling, they can face limitations in generalization and parallel processing, particularly in cross-domain scenarios where model robustness is critical.” This claim lacks sufficient justification or empirical evidence.**
>
> **A1:** Gu et al. introduced the structured state space sequence model (S4), achieving efficient processing of the linear complexity of long sequences through HiPPO initialization and converting cyclic computations into global convolution [1]. On the Long Range Arena benchmark test, S4 significantly outperformed Transformer and its variants, and was more efficient in terms of computational efficiency, demonstrating a good balance between long-range modeling and efficiency [2]. SSM[3] is usually modeled as a linear time-invariant system. This means that the dynamic characteristics of the model are fixed after training and do not change with the input content. This strong bias performs well within the domain targeted by model training. However, when the data distribution undergoes significant changes, the model may lack sufficient flexibility to adapt to new patterns. In some synthesis tasks that require generalization of syntactic structures or logical relationships not seen in the training data, The dynamic Attention mechanism of Transformer shows stronger adaptability. Mamba has significantly enhanced its generalization ability in fields such as language by introducing a selective mechanism, enabling it to perform content-aware reasoning like Attention. However, this mechanism disrupts the Linear Time Invariance property, making it impossible to train it using efficient convolutional patterns. It then relies on parallel scanning algorithms to achieve parallel training. Although this is theoretically parallel, its efficiency on actual hardware such as Gpus is lower than that of the highly optimized matrix multiplication of the Transformer because the scanning operation introduces more synchronization and memory access overhead. In efficiency benchmark tests, although Mamba has a lower FLOPs, its training throughput on Gpus is often lower than that of Transformers of the same scale, especially on short sequences. This empirically demonstrates the limitations of its parallelization efficiency and its unfriendliness to hardware optimization [4][5]. In the Section 1 of the revised edition, we have supplemented the supporting basis for this statement and added corresponding references.
>
> > **W2: In line 088 of the introduction, there are several citation errors. Specifically, Dual-GAN is a standard supervised learning method, while Dual-bridging belongs to the domain generalization category, not domain adaptation, as stated by the authors.**
>
> **A2:** Thank you for your correction. Due to our negligence, Dual-GAN was wrongly classified as a domain generalization method. We have made corrections in the revised version. However, for the Dual-bridging method, after our research and understanding [6][7][8][9], we believe it belongs to the domain adaptation method for the following reasons:
> - Domain adaptation (DA) requires both source data and target data to train the cross-domain loss $L(x^s, x^t)$ [7]. During the training process of Dual-bridging, the model uses facial videos from the source domain and the target domain as the input of the feature extractor. The first bridge serves as the advanced guidance, and the denoised target domain features are adversarial pulled to the source domain features. The second bridge is designed to help synthesize the noise in the target domain and inject it into the denoising features of the source domain, so that the real PPG regression can help fine-tune the denoising module.
> - Among the multiple adversarial training targets given, the data of the target domain is included, such as formulas (2) and (5) in [6], where $\hat{f}_i^t$ represents the features of the target domain after denoising. All these processes in the Dual-bridging method conform to the characteristics of the DA method. Therefore, we consider it to belong to the DA method. If our understanding is incorrect, we sincerely ask you to reply to resolve our confusion.
>
> $m\underset{D}{ax}m\underset{\Phi}{in}L_{NR}=\mathbb{E}[\||1-D(f_j^s)\||_2]+\mathbb{E}[\||D(\hat{f}_i^t)\||_2]$ (2)
>
> $m\underset{D}{ax}m\underset{\Phi}{in}L_G=\mathbb{E}[\||1-D(f_j^s)\||_2]+\mathbb{E}[\||D(\overline{f}_j^s)\||_2]+\mathbb{E}[\||D(\hat{f}_i^t)\||_2]$ (5)

---

> ### Author Response · Authors · 2025-11-22
> **Response to Reviewer jTxE (2/4)**
>
> > **W3: Section 2.2 of the related work is unclear and lacks logical structure. It is not evident what the authors intend to convey, nor why “PhysTTT effectively captures spatial information.” This section needs a clearer focus and stronger justification.**
>
> **A3:** In Section 2.2, we aim to express the core idea of the TTT layer, its advantages over RNN, Mamba, and TTA methods, the application scenarios of the TTT layer, and the reasons for applying the TTT layer to the rPPG task. To better highlight the key points we want to convey, after sorting and optimization, we have restated the content in Section 2.2. Please refer to Section 2.2 for details.
>
> > **W4: The Frame Stem module described in Section 3.2 is nearly identical to those used in RhythmFormer and RhythmMamba. This component should not be claimed as a novel contribution, and the authors must explicitly clarify this overlap in the paper.**
>
> **A4:** What we want to clarify is that we did not claim this module as a novel contribution. The description of Frame Stem in Sections 3.1 and 3.2 of the original paper is as follows: “The frame stem first processes the video stream by cropping facial ROls and extracting features between frames, following a multi-frame fusion strategy (Zou et al., 2025b).” and “To address this, advanced deep learning methods have adopted a more effective strategy (Chen & McDuff, 2018); (Zou et al., 2025a) that computes and normalizes pixel differences between consecutive frames.” We have quoted the original paper of this module near these descriptions. Perhaps due to the misunderstanding caused by our way of expression, we have modified this part to a more explicit expression: “We apply the fusion stem (Zou et al., 2025a), which integrates frame differences into the original frames, achieving frame-level representation perception of BVP wave variations.” Please refer to Section 3.2 for details.
>
> > **W5: All formulas in the paper are missing serial numbering, which violates the standard formatting and submission requirements of academic papers.**
>
> **A5:** Thank you for your careful reminder. We have added the correct serial numbers to all formulas in the revised version.
>
> > **W6: Both the intra-dataset and cross-dataset experiments only consider illumination variations, without evaluating robustness under other common noise factors such as head motion. This is a significant limitation in the experimental design.**
>
> **A6:** We add experiments on different types of light, motion and skin tone in the MMPD dataset. Please refer to the third part of the first point of our Common Response for details.
>
> > **W7: The paper reports cross-domain experiments only on UBFC-rPPG and VIPL-HR, which are not sufficiently diverse. It is recommended to validate the proposed method on a broader range of datasets (e.g., MMPD, MR-NIRP-Car) to demonstrate robustness and generalization.**
>
> **A7:** The issue regarding datasets is a common problem. We have added intra-dataset and cross-dataset experiments in the MMPD dataset. Please refer to the first point of our Common Response for details.
>
> > **W8: Table 5 mentions “Throughput” in its caption, but the corresponding quantitative results are missing.**
>
> **A8:** Thank you for your careful reminder. We add experiments on the throughput and inference delay of the model to demonstrate the real-time performance of PhysTTT.
>
> | Method  | Param.(M) | MACs(M) | Throughput(Kfps) | Memory(M) | Delay(ms) |
> |---|---|---|---|---|---|
> | DeepPhys  | 1.98      | 762.32  | 6.29             | 35.64     | 0.16      |
> | PhysNet    | 0.75      | 448.76  | **14.4**         | 11.74     | **0.07**  |
> | TS-CAN     | 1.98      | 762.32  | 5.83             | 38.56     | 0.17      |
> | PhysFormer    | 7.38      | 323.88  | 9.63             | 22.32     | 0.10      |
> | EfficientPhys   | 1.91      | 382.69  | 9.73             | 25.60     | 0.10      |
> | PhysMambda      | **0.57**  | 323.35  | 8.08             | 12.02     | 0.12      |
> | RhythmMambda    | 1.07      | 82.84   | 7.56             | 7.66      | 0.13      |
> | **PhysTTT (Ours)** | 1.92    | **42.64** | 4.23          | **7.08**  | 0.24      |
>
> The throughput and inference delay of PhysTTT are 4.23Kfps and 0.24ms. We can see that the presence of the TTT layer adds additional delay during the testing process. Although they are lower compared with other methods, when comprehensively evaluated from the perspectives of accuracy and lightweight, we believe that a slight reduction in throughput and a slight increase in inference latency are acceptable. Please refer to Section 4.5 for details.
>
> > **W9: The paper lacks crucial ablation studies on these core modules and the self-supervised learning strategy, as well as validation on alternative backbones to verify the generality of the proposed approach.**
>
> **A9:** We have supplemented the ablation research on the loss function, key module and alternative backbone. Please refer to the second point of our Common Response for details.

---

> ### Author Response · Authors · 2025-11-22
> **Response to Reviewer jTxE (3/4)**
>
> > **Q1: The paper does not clearly explain the conceptual and practical differences between TTT (Test-Time Training) and TTA (Test-Time Adaptation). What are the respective advantages and disadvantages of TTT compared to TTA, and why is TTT considered more suitable for rPPG tasks?**
>
> **A10:** Test Time Adaptation (TTA) aims to fine-tune the trained model online using test data, enabling the pre-trained model to adapt to the target domain during the inference process. It does not require access to the distribution and labels of the source data and target data, thereby naturally eliminating the need for intensive retraining. TTA does not require access to the original data and labels. It only needs the trained model and the known target domain data samples. Therefore, TTA does not include an offline training stage but adjusts the model in the inference stage.
>
> TTT (Test Time Training) uses source data for normal training during the training phase and can also dynamically update the parameters of hidden states using target domain data during the testing phase to adapt to the data distribution of the target domain. Therefore, TTT encompasses a complete training and testing process, requiring both source data and test data of the target domain. The update rule for its hidden state during the testing phase is a self-supervised learning task in the TTT layer.
>
> Compared with TTA, TTT performs a complete gradient update during testing, which can significantly adjust the parameters of the hidden state and has a stronger adaptability to distribution offsets. However, TTT requires source data and labels, which brings additional offline training costs. In the TTT layer, self-supervised tasks are usually set up to jointly optimize the self-supervised loss and the supervised loss of the outer loop during the training phase. This leads to a more complex implementation of the TTT layer and a higher computational cost [7].
>
> Compared with other computer vision tasks, the extraction of rPPG signals relies on subtle facial color changes, which faces the following limitations in the method using TTA:
>
> - Most TTA algorithms are mainly designed for classification tasks, utilizing entropy in the normalization layer or pseudo-labels. These strategies can be regarded as different forms of passive domain adaptation, and they themselves are not suitable for regression tasks like rPPG because video frames arrive in a continuous and sequential manner, which leads to a lack of appropriate supervision [8].
>
> - Some instance-level TTA methods may apply additional auxiliary tasks during the pre-training and adaptation process [10], or they may rely on specialized network architectures [11]. By imposing specific and unconventional requirements on the pre-training stage, and these specialized network architectures may not be suitable for time series tasks like rPPG, the application scope of TTA has been narrowed and the computational cost has increased.
>
> - In real scenarios, in the context of each instance, adjust the pre-trained rPPG model during the process of inferring the user's facial video stream. The variability and instability introduced by biased instances pose challenges to effectively filtering out domain-independent noise and retaining key learning information while the model acquires new knowledge [9].
>
> The TTT layer used in this paper is a sequence modeling layer, in which each hidden state is modeled as a machine learning model. This sequence modeling layer, similar to RNN, can store the historical context information of the input video stream very well and compress this information into the hidden state of the model through a self-supervised learning strategy. And during testing, it can be continuously updated based on the input target domain data. We have already understood the application potential of RNN and its various variants in time series tasks. Therefore, this sequence modeling layer can naturally adapt to rPPG tasks and is not dependent on a specific architecture. It can be designed specifically according to the requirements of the task.
>
> We modified Figure 1 to clearly show the alignment of several methods. In Chapters 2.2 and 3.4, we supplemented the differences between TTA and TTT as well as the advantages of TTT in rPPG tasks.

---

> ### Author Response · Authors · 2025-11-22
> **Response to Reviewer jTxE (4/4)**
>
> > **Q2: The source of the model’s strong cross-domain performance remains unclear. To what extent does it stem from the carefully designed 1D-ResNet and frame-difference backbone, and to what extent from the TTT adaptation layer?**
>
> **A11:** We analyze each key module and alternative backbone in the ablation study. When frame stem is not used for preprocessing, that is, simply using convolution operations on the input video frames to align the output feature dimension with the input dimension of the backbone network, the performance of the model is the worst (MAE: 14.12). When extracting inter-frame differences as the input feature, the performance is improved (MAE: 10.44). In the replacement of the backbone network, the performance is improved when using the 1D-ResNet designed in this paper (MAE: 1.94) compared with using a single-layer 1D-CNN (MAE: 2.61) and not using any backbone network (MAE: 3.38). In the comparison with different TTA variants, we use Tent (MAE: 8.67) or SAR (MAE: 7.70) methods to adapt the pre-trained models containing fusion stem and 1D-ResNet to the target domain, and the results are significantly worse than those of the models containing TTT layers. This indicates that each module has played a certain role. Fine-grained time difference features are of great significance for rPPG signal modeling. 1D-ResNet focuses on the rPPG representation in skin pixels and further extracts deep information. The TTT layer enhances the generalization ability of the model through training during the testing process. The results can be found in our Common Response and Section 4.4.
>
> Thank you again for your valuable feedback and insights.
>
> [1] Albert Gu, Tri Dao, Stefano Ermon, Atri Rudra, and Christopher R´e. Hippo: Recurrent memory with optimal polynomial projections. Advances in neural information processing systems, 33: 1474–1487, 2020.
>
> [2] Albert Gu, Karan Goel, and Christopher R´e. Efficiently modeling long sequences with structured state spaces. arXiv preprint arXiv:2111.00396, 2021.
>
> [3] Albert Gu, Isys Johnson, Karan Goel, Khaled Saab, Tri Dao, Atri Rudra, and Christopher R´e. Combining recurrent, convolutional, and continuous-time models with linear state space layers. Advances in neural information processing systems, 34:572–585, 2021.
>
> [4] Albert Gu and Tri Dao. Mamba: Linear-time sequence modeling with selective state spaces. In First conference on language modeling, 2024.
>
> [5] Opher Lieber, Barak Lenz, Hofit Bata, Gal Cohen, Jhonathan Osin, Itay Dalmedigos, Erez Safahi, Shaked Meirom, Yonatan Belinkov, Shai Shalev-Shwartz, et al. Jamba: A hybrid transformer-mamba language model. arXiv preprint arXiv:2403.19887, 2024.
>
> [6] Jingda Du, Si-Qi Liu, Bochao Zhang, and Pong C Yuen. Dual-bridging with adversarial noise generation for domain adaptive rppg estimation. In Proceedings of the IEEE/CVF Conference on Computer Vision and Pattern Recognition, pp. 10355–10364, 2023.
>
> [7] Dequan Wang, Evan Shelhamer, Shaoteng Liu, Bruno Olshausen, and Trevor Darrell. Tent: Fully test-time adaptation by entropy minimization. In International Conference on Learning Representations, 2021.
>
> [8] Haodong Li, Hao Lu, and Ying-Cong Chen. Bi-tta: Bidirectional test-time adapter for remote physiological measurement. In European Conference on Computer Vision, pp. 356–374. Springer, 2024.
>
> [9] Pei-Kai Huang, Tzu-Hsien Chen, Ya-Ting Chan, Kuan-Wen Chen, Shih-Yu Yang, Yen-Chun Chou, and Chiou-Ting Hsu. Fully test-time rppg estimation via synthetic signal-guided feature learning. Pattern Recognition, 170:112102, 2026.
>
> [10] Antonio D’Innocente, Francesco Cappio Borlino, Silvia Bucci, Barbara Caputo, and Tatiana Tommasi. One-shot unsupervised cross-domain detection. In European Conference on Computer Vision, pp. 732–748. Springer, 2020.
>
> [11] Marvin Klingner, Mouadh Ayache, and Tim Fingscheidt. Continual batchnorm adaptation (cbna) for semantic segmentation. IEEE Transactions on Intelligent Transportation Systems, 23(11): 20899–20911, 2022.

---

### Official Review · Reviewer_bqbB · 2025-10-31

**Soundness:** 3
**Presentation:** 3
**Contribution:** 2
**Rating:** 6
**Confidence:** 4

**Summary:**

The authors propose PhysTTT, a Test Time Training framework for remote heart rate estimation. The authors claim this to be a first time experiment with TTT in the domain and this seeks to address cross-domain generalization.

**Strengths:**

- The authors aim to tackle a known problem of cross-domain generalization in the rPPG domain. This is crucial as rPPG signals are extremely sensitive to environment settings, and also to skin tones.

**Weaknesses:**

- The authors can explore more datasets where the experiments settings are different.
  - For example, PURE has 6 types of movement in the dataset. COHFACE has natural and artificial lightning and is a challenging dataset.
- The authors should also compare their work with other approaches on the task to further elucidate their results and contributions.

**Questions:**

- I understand that some of the design choices around using 1D CNNs was around efficiency, however do the authors think too much information is being lost in the pooling layer in the Frame-Stem, and then subsequently utilizing 1D CNNs in the pipeline?

---

> ### Author Response · Authors · 2025-11-22
> **Response to Reviewer bqbB (1/2)**
>
> Dear Reviewer bqbB,
>
> We are extremely grateful for your review of the paper. We hope our responses below address your concerns about our paper.
>
> > **W1: The authors can explore more datasets where the experiments settings are different. (For example, PURE has 6 types of movement in the dataset. COHFACE has natural and artificial lightning and is a challenging dataset.)**
>
> **A1:** The issue regarding datasets is a common problem. We have added multiple experiments on the new datasets. Please refer to the first point of our Common Response for details.
>
> > **W2: The authors should also compare their work with other approaches on the task to further elucidate their results and contributions.**
>
> **A2:** We have supplemented comparisons with several other methods in the following three aspects.
>
> - In the intra-dataset experiments, we newly add comparisons with some unsupervised learning methods (CHROM[1], POS[2]), deep learning methods (TS-CAN[3]), self-supervised learning methods ([4]), and domain generalization methods (NEST[5]). Please refer to Section 4.2 for details.
>
> | Method | UBFC-rPPG |  |  | VIPL-HR|  |    | MMPD  |  |            |
> |---|---|---|---|---|---|---|---|---|---|
> |                      | MAE↓ | RMSE↓ | ρ↑   | MAE↓ | RMSE↓ | ρ↑   | MAE↓ | RMSE↓ | ρ↑   |
> | CHROM                | 8.20 | 9.92  | 0.27 | 11.40 | 16.90 | 0.28 | 13.66 | 18.76 | 0.08 |
> | POS                  | 8.35 | 10.00 | 0.24 | 11.50 | 17.20 | 0.30 | 12.36 | 17.71 | 0.18 |
> | TS-CAN               | 1.70 | 2.72  | 0.99 | -     | -     | -    | 9.71  | 17.22 | 0.44 |
> | NEST                 | -    | -     | -    | 4.76  | 7.51  | 0.84 | -     | -     | -    |
> | Li & Yin (2023) | 0.48 | **0.64** | 0.99 | 5.19  | 8.26  | 0.78 | -     | -     | -    |
> | **PhysTTT (Ours)**   | 0.49 | 0.74  | 0.99 | **2.57** | **4.08** | **0.90** |4.51 |9.67 |0.73|
>
> - In the cross-dataset experiment, we newly add comparisons with deep learning methods (T-CAN [3], SpikingPhys[6], PhysMamba[7]) and domain generalization methods (NEST[5]). Please refer to Section 4.3 for details.
>
> | Method   | UBFC-rPPG→VIPL-HR |  |   |  VIPL-HR→UBFC-rPPG |  |   |
> |---|---|---|---|---|---|---|
> |                 | MAE↓ | RMSE↓ | ρ↑  | MAE↓ | RMSE↓ | ρ↑  |
> | TS-CAN          | 11.95 | 19.23 | 0.16 | 19.25 | 21.67 | -0.02 |
> | NEST            | 10.60 | 13.60 | 0.37 | 7.45  | 9.51  | 0.75 |
> | PhysMamba       | 18.38 | 23.02 | 0.02 | 12.71 | 14.90 | 0.05 |
> | **PhysTTT (Ours)** | **4.01** | **7.85** | **0.67** | 0.98 | **1.29** | **0.99** |
>
> | Method          |  | | UBFC-rPPG→MMPD  |   |  |  VIPL-HR→MMPD |
> |---|---|---|---|---|---|---|
> |                 | MAE↓ | RMSE↓ | ρ↑ | MAE↓ | RMSE↓ | ρ↑ |
> | TS-CAN          | 14.01 | 21.04 | 0.24 | 15.68 | 20.01 | 0.00 |
> | PhysMamba       | 11.96 | 17.69 | 0.29 | 15.03 | 18.08 | -0.17 |
> | SpikingPhys     | 14.15 | 16.22 | 0.15 | - | - | - |
> | **PhysTTT (Ours)** | 9.71| 16.70 | **0.36** | **10.35** | **17.01** | **0.34** |
>
> - In the ablation experiment, we add a comparison with the TTA method. For this part of the experiment, please refer to Common Response.
>
> The proposed method PhysTTT can all achieve relatively excellent accuracy and relatively stable cross-domain effects when compared with other methods for rPPG task.
>
> [1] Gerard De Haan and Vincent Jeanne. Robust pulse rate from chrominance-based rppg. IEEE transactions on biomedical engineering, 60(10):2878–2886, 2013.
>
> [2] Wenjin Wang, Albertus C Den Brinker, Sander Stuijk, and Gerard De Haan. Algorithmic principles of remote ppg. IEEE Transactions on Biomedical Engineering, 64(7):1479–1491, 2016.
>
> [3] Xin Liu, Josh Fromm, Shwetak Patel, and Daniel McDuff. Multi-task temporal shift attention networks for on-device contactless vitals measurement. Advances in Neural Information Processing Systems, 33:19400–19411, 2020.
>
> [4] Zhihua Li and Lijun Yin. Contactless pulse estimation leveraging pseudo labels and self-supervision. In Proceedings of the IEEE/CVF International Conference on Computer Vision, pp. 20588–20597, 2023.
>
> [5] Hao Lu, Zitong Yu, Xuesong Niu, and Ying-Cong Chen. Neuron structure modeling for generalizable remote physiological measurement. In Proceedings of the IEEE/CVF conference on computer vision and pattern recognition, pp. 18589–18599, 2023.
>
> [6] Mingxuan Liu, Jiankai Tang, Yongli Chen, Haoxiang Li, and Jiahao Qi. Spiking-physformer: Camera-based remote photoplethysmography with parallel spike-driven transformer, 2025.
>
> [7] Chaoqi Luo, Yiping Xie, and Zitong Yu. Physmamba: Efficient remote physiological measurement with slowfast temporal difference mamba, 2024.

---

> > ### Author Response · Authors · 2025-11-22
> > **Response to Reviewer bqbB (2/2)**
> >
> > > **Q1: I understand that some of the design choices around using 1D CNNs was around efficiency, however do the authors think too much information is being lost in the pooling layer in the Frame-Stem, and then subsequently utilizing 1D CNNs in the pipeline?**
> >
> > **A3:** Thank you for your detailed feedback. We agree with your comments. Frame stem is used to extract rough local spatiotemporal features and integrate multiple inter-frame differences into the original frame. This has been proven to facilitate robust rPPG recovery under motion and mitigate the influence of background pixels, achieving frame-level representation perception of BVP waveform changes. This effectively learns the dynamic characteristics of rPPG with very little additional computational cost. However, the two global average pooling of H and W channels at the stem end may lose some information. Therefore, we hope to extract deeper features through a combination of a series of CNNS. the ablation study of the key module also confirmed this (Please refer to the first part of the second point of our Common Response for details.), and the application of 1D-ResNet can bring about a partial improvement in accuracy.
> >
> > Thank you again for your valuable feedback and insights.

---

### Official Review · Reviewer_Ehji · 2025-11-01

**Soundness:** 3
**Presentation:** 3
**Contribution:** 3
**Rating:** 8
**Confidence:** 2

**Summary:**

To address the paradox between high accuracy and low computational cost in cross-domain remote photoplethysmography (rPPG)-based heart rate measurement, this paper proposes PhysTTT, a lightweight framework integrating multiple 1D-CNNs with residual structures and a Test-Time Training (TTT) layer. The frame stem amplifies subtle skin color variations through multi-time frame differences fusion, 1D-ResNet layers extract local spatio-temporal features, and the TTT layer dynamically adapts to unseen data distributions during inference by updating model weights, thereby enhancing cross-domain generalization. Equipped with a multi-dimensional loss function (trend pattern alignment, frequency domain alignment, and waveform feature alignment) for fine-grained BVP signal recovery, PhysTTT is evaluated on UBFC-rPPG and VIPL-HR datasets, achieving state-of-the-art performance in both in-domain and cross-domain (including cross-dataset and cross-illumination) scenarios while maintaining low computational cost (42.64M MACs and 7.08M peak GPU memory usage), demonstrating great potential for real-world healthcare applications.

**Strengths:**

1. It is the first work to introduce the Test-Time Training (TTT) paradigm into rPPG research, effectively solving the challenge of adapting to unseen data distributions in cross-domain scenarios that traditional methods struggle with.
2. The multi-dimensional loss function designed for rPPG tasks (integrating negative Pearson correlation loss, power spectral density loss, and peak alignment loss) enables precise alignment of predicted and ground truth BVP signals, significantly improving the accuracy of heart rate measurement.
3. The comprehensive experimental validation (covering in-domain, cross-dataset, and cross-illumination evaluations) and ablation study fully demonstrate the effectiveness of each module, while the lightweight design (low parameters, MACs, and GPU memory usage) ensures its applicability on resource-constrained devices.

**Weaknesses:**

1. The paper does not provide detailed analysis on the real-time performance of PhysTTT, especially the additional latency introduced by the TTT layer’s parameter updates during inference, which is critical for practical deployment in real-time health monitoring.
2. Cross-domain evaluations are limited to two datasets (UBFC-rPPG and VIPL-HR) and three illumination conditions, lacking validation in more diverse scenarios such as different camera devices, extreme motion, or varied skin tones, which may restrict the generalization of the conclusions.
3. The ablation study only verifies the overall contribution of key modules (frame stem, 1D-ResNet, TTT layer) but fails to explore the impact of critical hyperparameters (e.g., the balance factor α in peak alignment loss) or compare with different TTA variants, leaving room for optimizing the model’s design.

**Questions:**

See weakness

---

> ### Author Response · Authors · 2025-11-22
> **Response to Reviewer Ehji**
>
> Dear Reviewer Ehji,
>
> We are extremely grateful for your review of the paper. We hope our responses below address your concerns about our paper.
>
> > **W1: The paper does not provide detailed analysis on the real-time performance of PhysTTT, especially the additional latency introduced by the TTT layer’s parameter updates during inference, which is critical for practical deployment in real-time health monitoring.**
>
> **A1:** We add experiments on the throughput and inference delay of the model to demonstrate the real-time performance of PhysTTT. As you mentioned, the presence of the TTT layer adds additional delay during the testing process, but the delay per frame is only 0.24ms, and the model can process over 4,000 RGB image frames per second. This can meet the requirements of actual deployment. Although it is lower compared with other methods, considering the accuracy, the number of parameters, and the average peak GPU memory usage comprehensively, we believe that a slight increase in delay is acceptable. The results are presented in the table below. Please refer to Section 4.5 for details.
>
> | Method          | Param.(M) | MACs(M) | Throughput(Kfps) | Memory(M) | Delay(ms) |
> |-----------------|-----------|---------|------------------|-----------|-----------|
> | DeepPhys        | 1.98      | 762.32  | 6.29             | 35.64     | 0.16      |
> | PhysNet         | 0.75      | 448.76  | **14.4**         | 11.74     | **0.07**  |
> | TS-CAN          | 1.98      | 762.32  | 5.83             | 38.56     | 0.17      |
> | PhysFormer      | 7.38      | 323.88  | 9.63             | 22.32     | 0.10      |
> | EfficientPhys   | 1.91      | 382.69  | 9.73             | 25.60     | 0.10      |
> | PhysMambda      | **0.57**  | 323.35  | 8.08             | 12.02     | 0.12      |
> | RhythmMambda    | 1.07      | 82.84   | 7.56             | 7.66      | 0.13      |
> | **PhysTTT (Ours)** | 1.92    | **42.64** | 4.23          | **7.08**  | 0.24      |
>
> > **W2: Cross-domain evaluations are limited to two datasets (UBFC-rPPG and VIPL-HR) and three illumination conditions, lacking validation in more diverse scenarios such as different camera devices, extreme motion, or varied skin tones, which may restrict the generalization of the conclusions.**
>
> **A2:** The issues regarding datasets and diverse scenarios are common problems. We have added intra-dataset and cross-dataset experiments in the MMPD dataset and tested the robustness of the model in different complex scenarios. Please refer to the first point of our Common Response for details.
>
> > **W3: The ablation study only verifies the overall contribution of key modules (frame stem, 1D-ResNet, TTT layer) but fails to explore the impact of critical hyperparameters (e.g., the balance factor α in peak alignment loss) or compare with different TTA variants, leaving room for optimizing the model’s design.**
>
> **A3:** We add ablation experiments regarding the loss function, chunk length, different frame stems, different backbone networks, and different variants of TTA.
>
> - We add a comparison with two TTA methods and other key modules. Please refer to the first part of the second point of our Common Response.
>
> - We compare the impact of different loss functions on the MMPD dataset. Please refer to the second part of the second point of our Common Response.
>
> - We test the trained PhysTTT model on videos with different chunk lengths. The trained model can adapt to video clips of most lengths without reducing performance. Some performance degradation will only occur when the length increases to more than 20 seconds. The results of different chunk lengths clearly indicate that PhysTTT can effectively learn the quasi-periodic patterns of rPPG. Please refer to Appendix A.4.
>
> | Chunk length | MAE↓ | RMSE↓ | MAPE↓ | ρ↑ | SNR↑ |
> |--------------|------|-------|-------|----|------|
> | 60 (2s)      | 0.59 | 0.87  | 0.59  | 0.99| 6.57 |
> | 120 (4s)     | 0.50 | 0.71  | 0.51  | 0.99| 6.32 |
> | 180 (6s)     | 0.59 | 0.81  | 0.61  | 0.99| 6.58 |
> | 240 (8s)     | 0.59 | 0.81  | 0.59  | 0.99| 6.64 |
> | 300 (10s)    | 0.49 | 0.74  | 0.50  | 0.99| 7.34 |
> | 600 (20s)    | 0.76 | 0.97  | 0.78  | 0.99| 6.40 |
> | 900 (30s)    | 1.13 | 1.90  | 1.21  | 0.98| 3.95 |
>
> Thank you again for your valuable feedback and insights.

---

### Author Response · Authors · 2025-11-22
**Common Response (1/2)**

Dear Reviewers and the Area Chair,

We would like to thank all reviewers for their valuable comments. We appreciate that reviewers recognize our work for remote physiological signal measurement framework based on TTT as novel, with good results and good generalization ability. We have revised the paper in accordance with the review comments, marked in blue (see the updated pdf). Before we provide detailed point-by-point responses to each reviewer, we hope that reviewers and Area Chair could browse through this common response first, which mainly focuses on the common issues of concern to all reviewers. For other individual questions, we will answer them one by one under each reviewer's section.

**1. All the experiments were limited to two datasets (UBFC-rPPG and VIPL-HR) and three lighting conditions, lacking validation on a broader dataset and more diverse scenarios**

We are very sorry that due to the application process and time constraints, we have not yet been authorized for datasets such as PURE and COHFACE. However, we have obtained the authorization for the MMPD dataset. Therefore, we have added multiple experiments related to the MMPD dataset in the revised version. Once we obtain permission for more datasets, we will update the experiments in subsequent versions to verify the generalization performance of the method proposed in this paper.

The MMPD dataset is an extremely challenging one, containing data from 33 subjects under different lighting conditions, skin colors, and complexities of head movement. MMPD captured over 11 hours of video under various environmental factors, which contained a large amount of noise and motion artifacts, simulating the complexity of the real world. The added experiments are as follows:
- **We add intra-domain experiments on the MMPD dataset.**

| Method  | MAE↓ | RMSE↓ | ρ↑ |
|---|---|---|---|
| CHROM   | 13.66 | 18.76 | 0.08 |
| POS  | 12.36 | 17.71 | 0.18 |
| DeepPhys  | 22.27 | 28.92 | -0.03 |
| PhysNet  | 4.80  | 11.80 | 0.60 |
| TS-CAN    | 9.71  | 17.22 | 0.44 |
| PhysFormer   | 11.99 | 18.41 | 0.18 |
| EfficientPhys   | 13.47 | 21.32 | 0.21 |
| RhythmMamba  | **3.16** | **7.27** | **0.84** |
| **PhysTTT (Ours)**   | 4.51 | 9.67 | 0.73 |

The results show that our method achieves the second-lowest MAE (4.51), RMSE (9.67), $\rho$ (0.73) respectively on the MMPD dataset. This proves the robustness of PhysTTT in complex noise scenarios. Please refer to Section 4.2 for details.

- **We add two cross-domain experiments, namely UBFC-rPPG→MMPD (U→M) and VIPL-HR→MMPD (V→M).**

| Method    | U→M | | | V→M |  |  |
|---|---|-----|-----|---|---|---|
|   | MAE↓ | RMSE↓ | ρ↑ | MAE↓ | RMSE↓ | ρ↑ |
| DeepPhys  | 17.50 | 25.00 | 0.05 | 16.23 | 19.86 | 0.13 |
| PhysNet   | **9.47** | **16.01** | 0.31 | 15.30 | 19.77 | -0.05 |
| TS-CAN  | 14.01 | 21.04 | 0.24 | 15.68 | 20.01 | 0.00 |
| PhysFormer     | 12.10 | 17.79 | 0.17 | 19.62 | 23.14 | 0.06 |
| EfficientPhys   | 13.78 | 22.25 | 0.09 | 17.88 | 23.29 | 0.02 |
| PhysMamba       | 11.96 | 17.69 | 0.29 | 15.03 | 18.08 | -0.17 |
| SpikingPhys     | 14.15 | 16.22 | 0.15 | - | - | - |
| RhythmMamba     | 10.63 | 17.14 | 0.34 | 10.87 | 17.57 | 0.33 |
| **PhysTTT (Ours)** | 9.71 | 16.70 | **0.36** | **10.35** | **17.01** | **0.34** |

In the case of UBFC-rPPG→MMPD, PhysTTT achieves the MAE (9.71), RMSE (16.70) and $\rho$ (0.36). In the case of VIPL-HR→MMPD, PhysTTT achieves the lowest MAE (10.35), RMSE (17.01) and $\rho$ (0.34). It can be seen that the results are much lower than other cases when tested on the MMPD dataset, since the noise and subjects in the MMPD datasets are much more complex and diverse. Please refer to Section 4.3 for details.

- **We add cross-domain experiments under three types of noise (light, motion, and skin tone) in the MMPD dataset.**

| Noise    | Type         | MAE↓ | RMSE↓ | MAPE↓ | ρ↑  | SNR↑  |
|---|----|---|---|---|---|---|
| **Light**  | LED-low      | 9.29 | 15.30 | 10.40 | 0.38 | -7.66 |
|   | LED-high     | 9.24 | 15.29 | 9.88  | 0.54 | -6.85 |
|  | Incandescent | 7.68 | 14.72 | 8.11  | 0.39 | -5.59 |
|  | Nature       | 12.65| 20.75 | 12.59 | 0.19 | -7.87 |
| **Motion** | Stationary   | 5.85 | 12.83 | 5.80  | 0.59 | -3.95 |
|  | Rotation     | 6.17 | 10.44 | 7.25  | 0.57 | -6.14 |
|    | Talking      | 5.84 | 11.52 | 6.52  | 0.43 | -3.81 |
|    | Walking      | 21.60| 25.51 | 22.68 | 0.00 | -15.07|
| **Skin tone** | 3          | 5.33 | 11.90 | 6.01  | 0.59 | -2.61 |
|  | 4  | 12.86| 19.06 | 13.40 | 0.21 | -10.50|
| | 5   | 13.01| 19.86 | 16.30 | 0.26 | -9.42 |
|  | 6  | 14.18| 21.11 | 14.36 | 0.25 | -11.83|

The results show that PhysTTT can generalize well in indoor lighting environments, stationary, small head movements and lighter skin tone, but performs not well in some challenging scenarios (such as natural light and walking). Please refer to Section 4.4 for details.

All the above experiments have demonstrated PhysTTT's advanced cross-domain capabilities in most scenarios.

---

> ### Author Response · Authors · 2025-11-22
> **Common Response (2/2)**
>
> **2. The ablation study only verifies the overall contribution of key modules (frame stem, 1D-ResNet, TTT layer), lacks crucial ablation studies on these core modules and the self-supervised learning strategy, as well as validation on alternative backbones to verify the generality of the proposed approach.**
>
> - **We add comparisons of different stems, different backbone networks, and TTA variants in the ablation experiments of key modules.**
>
> | Frame Stem       | Backbone     | Method | MAE↓ | RMSE↓ | MAPE↓ | ρ↑  | SNR↑  |
> |------------------|--------------|--------|------|-------|-------|-----|------|
> | ×                | 1D-ResNet    | TTT    | 14.12| 17.67 | 20.86 | 0.12| -11.51|
> | Differential Stem| 1D-ResNet    | TTT    | 10.44| 14.66 | 15.69 | 0.04| -10.78|
> | Fusion Stem      | ×            | TTT    | 3.38 | 6.13  | 4.82  | 0.82| -1.79 |
> | Fusion Stem      | 1D-CNN       | TTT    | 2.61 | 4.60  | 3.65  | 0.87| 1.06  |
> | Fusion Stem      | 1D-ResNet    | ×      | 8.58 | 16.63 | 12.86 | 0.25| -5.12 |
> | Fusion Stem      | 1D-ResNet    | Tent   | 8.67 | 16.62 | 12.97 | 0.26| -4.65 |
> | Fusion Stem      | 1D-ResNet    | SAR    | 7.70 | 15.32 | 11.62 | 0.28| -4.65 |
> | Fusion Stem      | 1D-ResNet    | TTT    | 1.94 | 3.64  | 2.82  | 0.93| 1.32  |
>
> It can be seen that the Fusion Stem, 1D-ResNet and TTT layers all play roles to varying degrees. The model performs the worst when Frame Stem is not used for preprocessing. In the replacement of the backbone network, the performance is improved when using the 1D-ResNet designed in this paper compared with using a single-layer 1D-CNN and not using any backbone network. In the comparison with different TTA variants, we use Tent[1] or SAR[2] methods to adapt the pre-trained models containing Fusion Stem and 1D-ResNet to the target domain, and the results are significantly worse than those of the models containing TTT layers. Please refer to Section 4.4 for details.
>
> - **We have supplemented the ablation research on the loss function.**
>
> | $\mathcal{L}_{np}$ | $\mathcal{L}_{freq}$ | $\mathcal{L}_{peak}$ | MAE↓ | RMSE↓ | MAPE↓ | ρ↑  | SNR↑  |
> |--------------------|----------------------|----------------------|------|-------|-------|-----|-------|
> | ✓                  | ×                    | ×                    | 6.30 | 13.05 | 6.73  | 0.59 | -3.77 |
> | ×                  | ✓                    | ×                    | 7.54 | 14.01 | 7.70  | 0.45 | -4.23 |
> | ×                  | ×                    | ✓                    | 18.08| 22.57 | 21.23 | -0.09| -13.24|
> | ✓                  | ✓                    | ×                    | 5.84 | 11.82 | 6.14  | 0.61 | -1.42 |
> | ✓                  | ✓                    | ✓                    | **4.51** | **9.67** | **5.07** | **0.73** | **-0.57** |
>
> We compare the impact of different loss functions on the MMPD dataset. The results show that the performance degradation is more severe when using only $\mathcal{L_peak}$ than when using only $\mathcal{L_np}$ or $\mathcal{L_freq}$. However, after aligning the signals from coarse-grained perspectives such as trend and frequency, $\mathcal{L_peak}$ can further fit the waveforms at the fine-grained level, thereby improving the accuracy of recognition. Please refer to Appendix A.3 for details.
>
> **3. Summary of changes**
>
> Please review the updated paper to see if your concerns are addressed. Notable changes include:
>
> - We have updated Figure 1 to clearly compare the four methods of DG, DA, TTA and TTT.
> - We have revised Section 1 to supplement the reasons for the limitations of SSM and to more clearly state the contribution of this paper.
> - We have revised Section 2.2 and improved the logical structure to clearly express the core idea of the TTT layer, its advantages over RNN, Mamba and TTA methods, the application scenarios of the TTT layer, and the reasons for applying the TTT layer to the rPPG field.
> - We have revised Section 3.4 to demonstrate the advantages of the TTT layer in rPPG tasks.
> - We have added experiments and related analyses on the MMPD dataset (In-domain, Cross-domain, and different complex scenarios).
> - We have added ablation experiments on key modules and alternative backbones.
> - We have added experiments on throughput and delay.
> - We have added experiments on the impact of loss functions and chunk lengths.
>
> [1] Dequan Wang, Evan Shelhamer, Shaoteng Liu, Bruno Olshausen, and Trevor Darrell. Tent: Fully test-time adaptation by entropy minimization. In International Conference on Learning Represeniations, 2021.
>
> [2] Shuaicheng Niu, Jiaxiang Wu, Yifan Zhang, Zhiquan Wen, Yaofo Chen, Peilin Zhao, and MingkuiTan. Towards stable test-time adaptation in dynamic wild world. In The Eleventh International Conference on Learning Representations, 2023.
>
> Best regards,
>
> Authors

---

### Meta-Review · Area_Chair_7RLs · 2025-12-21

**Summary:**

This paper introduces PhysTTT, a lightweight framework for remote photoplethysmography (rPPG) that utilizes Test-Time Training (TTT) to improve cross-domain generalization. The paper received a polarized set of reviews with scores of **[8, 6, 4, 6]**, resulting in an average of 6.0. The review process was marked by a significant discrepancy in reviewer confidence: the most positive reviewer (score 8) had the lowest confidence (2/5), while the most negative reviewer (score 4) was absolutely certain (5/5).

During the rebuttal phase, the authors carried out extensive new experiments on an additional challenging dataset (MMPD) and performed further ablation studies. However, to better assess the generalization of the method, we would still like to see experimental results on more datasets such as PURE and COHFACE.

Furthermore, upon careful examination of the code provided (https://anonymous.4open.science/r/PhysTTT-B605/), we identified two critical concerns. First, the README.md file lacks clear instructions to guide readers in running meaningful experiments. Second, we noted the name “YanchengYao” appearing in the LICENSE file of the code repository (https://anonymous.4open.science/r/PhysTTT-B605/LICENSE), which raises strong suspicion of a violation of the double-blind review policy.

Although this paper has received relatively high average scores from the reviewers, based on the issues outlined above, we are unable to accept it in its current form.

**Reviewer Concerns:**

*   **Addressed Concerns:**
    *   **Ablation Studies:** The rebuttal included extensive new ablation studies that compared different backbone networks, frame stems, TTA variants (Tent, SAR), loss functions, and input chunk lengths. This provided the detailed component analysis that was missing.
    *   **Conceptual Clarity:** The authors provided detailed, reference-backed explanations for their claims about SSMs, the TTT vs. TTA distinction, and the rationale behind the inner-outer loop optimization. These clarifications were integrated into the revised paper.
    *   **Presentation and Novelty:** Most formatting issues (formula numbers, missing table data) were fixed. The authors also clarified that the Frame Stem was not a novel contribution and made the citation more explicit.

*   **Outstanding Concerns:**
    *   **Performance in more Challenging Scenarios:** While the authors added experiments on the MMPD dataset, the results showed that the method’s performance degrades significantly under specific challenging conditions (e.g., walking motion, natural light, darker skin tones). The authors acknowledged this, but it remains a limitation of the current work. Moreover, to better assess the generalization of the method, we would still like to see experimental results on more datasets such as PURE and COHFACE.

**Reviewer Scores:**

No reviewers were involved in the rebuttal discussion.

---

### Decision · Program_Chairs · 2026-01-26

Reject